# Generalizable and Actionable Parts Pose Estimation with Symmetry Annotation-Free Learning Strategy

Wenxiao Chen [*1]  Xueyu Yuan [*1]  Liu Liu [1]  Di Wu [2]  Dan Guo [1]

## Abstract

Urgently needed generalizable robot object interaction and manipulation requires high-quality Cross-Category object perception. As a pioneer of this area, Generalizable and Actionable Parts (GAParts) understanding has attracted increasing attention from relevant researchers. However, most recent works either have insufficient design regarding the symmetry issue or require rich symmetry annotation, which severely impedes precise GAPart pose estimation in data-lacking scenarios. In this paper, we propose SAFAG, a novel **S**ymmetry **A**nnotation-**F**ree framework for **G**eneralizable and **A**ctionable Parts Pose Estimation. Specifically, we suggest a stepwise refinement two-stage framework for candidate-to-final quaternion regression, and tackle the symmetry prediction as a probability distribution problem with self-supervised learning strategy. The experimental results demonstrate the superior performance and robustness of our SAFAG. We believe that our work has the enormous potential to be applied in many areas of embodied AI system.

## 1. Introduction

Model-based generalizable manipulation and interaction is a fundamental task in embodied AI research. Recent studies focusing on 6D pose estimation are pursuing the target of achieving the cross-category generalizability. Based on this, some works proposed to build up the pose estimation task on parts rather than the whole object. This is motivated by the following reasons. First, agents and robots always interact with the objects' parts instead of the entire objects. Second, parts classes are more elementary and fundamental compared to object categories. Geng *et al.* introduce the concept of Generalizable and Actionable Parts (GAParts)(Geng et al., 2023) to achieve the cross-category perception. However, since GAParts typically exhibit richer symmetry than entire objects, current methods always face two severe challenges while handling this. 1)**Multi-solution problem** GAPartNet introduces NPCS to solve for the unique pose solution which reduces the pose estimation accuracy. Because for objects and parts with symmetry, the ground truth is not the only feasible solution, but the entire equivalent set (*e.g.* rotating the slider lid of a kitchen pot by 180° around its symmetry axis provides a similar manipulation affordance) derived from ground truth annotation is. 2)**Annotation dependency in symmetry problem.** GASEM(Liu et al., 2025) proposes a modified framework by utilizing the point cloud and its motion information. Furthermore, DFGAP(Yuan et al., 2025) proposes an uncertainty-quantified method under depth-free condition. However, these works always design a set of symmetry-aware loss functions which highly rely on rich symmetry information and annotations (*e.g.* symmetry plane or axis). Since abundant and precise symmetry annotations are usually difficult to obtain, these methods' generalizability and practicality are hindered seriously without doubt.

In this paper, we propose SAFAG, a novel framework with **S**ymmetry **A**nnotation-**F**ree Learning Strategy for **G**eneralizable and **A**ctionable Parts Pose Estimation. Overall, we suggest a stepwise refinement two-stage framework with candidates generation and final quaternion aggregation. As for symmetry issue, we design an implicit and self-supervised learning strategy to robustly predict the symmetry axis/plane without any annotation of ground truth symmetry axis/plane. Specifically, with the input partial point cloud, we first leverage a modified 3DGCN backbone with tailored convolution design for better learning on the $S^3$ hyper-spherical manifold to extract the point cloud features. Our stepwise refinement framework consists of two stages. In the first stage, we generate a group of quaternion candidates around the ground truth quaternion. Right after the first stage is a small encoder, which aims to learn the implicit distribution information of the candidates and encode them with the point cloud features as the input of the second stage. Then in the second stage, we predict a

---

*Equal contribution [1]Hefei University of Technology, Hefei, Anhui, China [2]University of Science and Technology of China, Hefei, Anhui, China. Correspondence to: Liu Liu <liuliu@hfut.edu.cn>, Dan Guo <guodan@hfut.edu.cn>.

*Proceedings of the $43^{rd}$ International Conference on Machine Learning*, Seoul, South Korea. PMLR 306, 2026. Copyright 2026 by the author(s).

corresponding offset for each candidate in the tangent space of the $S^3$ hyper-spherical manifold. Finally, we employ a small CNN to aggregate them to obtain the final quaternion. In terms of symmetry, we estimate the symmetry axis or plane by modeling the probability distribution in $x, y, z$-axis. The final predicted axis or plane can be synthesized by the distribution density and weight. The entire process of symmetry learning is completely self-supervised, under the condition of missing annotation of symmetry axis or plane. With the predicted axis and plane, we can easily construct a symmetric equivalent set of ground truth, which can be applied to supervise the learning of the final quaternion.

We evaluate our SAFAG on the GAPartNet dataset, which is derived from two well-known benchmarks PartNet-Mobility(Xiang et al., 2020) and AKB-48(Liu et al., 2022). We also provide additional real-world experiment to evaluate the generalizability of method in the real world. Extensive qualitative and quantitative experiments show the superior performance of SAFAG in GAPart pose estimation task. Based on the high-quality GAPart perception results, we have adopted the interaction policies from GAPartNet to guide the generation of the final grasping action. The demonstrations illustrate that with the help of our SAFAG, the robot agent can achieve generalizable object manipulation task in the real world. **In summary**, our contributions can be concluded as follows:

1) We propose a novel framework SAFAG, making a further step in GAPart pose estimation task, which has tremendous potential for downstream robot-part interaction.

2) Our SAFAG achieves symmetry-annotation-free 6D pose estimation for rotational and mirror symmetry part, which enhances the generalizability in multiple data-scarce scenarios.

## 2. Related Works

### 2.1. Cross-Category Pose Estimation.

Instance-level (Xiang et al., 2017; Peng et al., 2019; Song et al., 2020; Schult et al., 2023; Chen et al., 2022) and category-level (Wang et al., 2019; Liu et al., 2023; Chen et al., 2024; Zhang et al., 2025; Chen & Dou, 2021; Lin et al., 2022; Li et al., 2020; Zhang et al., 2024b) pose estimation both aim to recover the 6D pose of objects, and both tasks have been extensively studied. Recent studies increasingly focus on cross-category pose estimation, which seeks to generalize pose estimation to unseen object categories during training. SAM-6D(Lin et al., 2024) treats pose estimation as a partial-to-partial point matching problem, where the matching process entirely relies on geometric consistency rather than category. Geng *et al.* propose the concept of Generalizable and Actionable Parts (GAParts)(Geng et al., 2023), upon which they develop simple heuristics that en-

able cross-category object pose estimation and manipulation. CAPE(Xu et al., 2022) reframes pose estimation as a keypoint matching task, aiming to create a model capable of detecting the pose of any class of object given only a few samples.

### 2.2. Symmetry-Induced Multi-hypothesis.

Regression-based pose estimation(Wang et al., 2019; Lin et al., 2021; Di et al., 2022; Chen et al., 2020; Zhang et al.; Lin et al., 2022; Chen et al., 2021; Lin et al., 2023; Chen et al., 2024) inherently possesses multi-hypothesis issue induced by object symmetries. Since the network is supervised with only a single ground-truth pose while multiple poses are feasible, the learning process becomes ill-posed and challenging. To tackle this problem, SARNet(Lin et al., 2022) and GPVPose(Di et al., 2022) utilize symmetry-aware reconstruction that predicts symmetrical point cloud $P'$ to extract more effective per-point features. SGPA(Chen & Dou, 2021) and NOCS(Wang et al., 2019) augment the ground truth pose for symmetric objects through the prior annotation of symmetry information. More recently, some studies(Ouyang et al., 2026; Zhang et al., 2024a; Xu et al., 2024; Zhang et al., 2023) introduce generative approaches to model the inherent one-to-many mapping. Zhang *et al.* propose GenPose(Zhang et al., 2023), which uses score-based diffusion models stochastically sampling all possible pose hypotheses as candidates. Ouyang *et al.* propose RFMPose(Ouyang et al., 2026), a novel Riemannian flow matching framework that leverages Riemannian Optimal Transport to find the shortest pose trajectories. However, these methods depend on annotations of symmetry information (*e.g.* symmetry axes or planes), which are often costly to obtain and thus restrict their practicality in applications.

## 3. Method

### 3.1. Overview

Following the definition of GAPart in GAPartNet(Geng et al., 2023), the problem formulation and notation are as follows: Given a partial point cloud $P = \{p_i\}_{i=1}^N \in \mathbb{R}^{N \times 3}$ of observed object, where $N$ denotes the number of points and is set to 1024 in our experiments. We assume that the object contains $L$ GAParts, with part label ranging from $\{1, \ldots, L\}$ and the $i$-th part has a class label $c_i \in \{1, \ldots, 9\}$. Our goal is to predict the 6-DoF pose of each individual GAPart, including 3-DoF rotation $R \in SO(3)$ and 3-DoF translation $t \in \mathbb{R}^3$. The overview of our framework is shown in Fig. 1. Specifically, the introduction of our framework is consists of four main parts: 1) HyperS3 convolution for quaternion regression; 2) quaternion regression with candidates to final; and 3) self-adaptive symmetry-aware design as well as 4) training strategy and loss function.

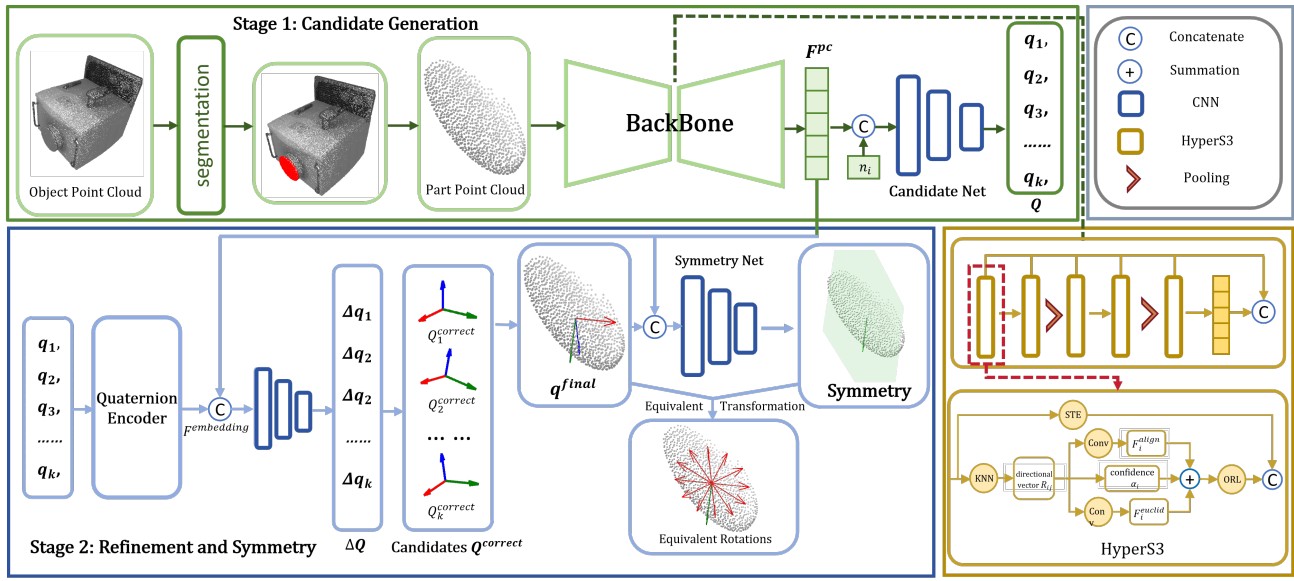

*Figure 1.* **Overview of our framework.** First, we construct a backbone with our designed $S^3$ hyperspherical (HyperS3) layer to extract point cloud feature. Then, we generate quaternion candidates and refine each on the hyperspherical manifold of quaternion $S^3$. To better cope with the multi-hypothesis caused by symmetry, we additionally design a self-adaptive network to estimate the symmetry axes or planes, based on which we generate the corresponding equivalent solutions later. Finally, we aggregate all refined candidates and apply an additional refinement step, yielding $q_{final}$.

### 3.2. HyperS3 convolution for Quaternion Regression

Recent works prove that modifying the fundamental backbone according to different specific task can robustly enhance the framework's performance.(Lin et al., 2023; Weijler & Hermosilla, 2025) Therefore, we propose HyperS3 convolution layer built specifically for quaternion regression by encouraging rotation-aware representations to remain consistent with the hyperspherical manifold $S^3$.

Given an input point cloud $P = \{p_i\}_{i=1}^N$, we first identify its $M$ nearest neighbors using a KNN search for each point, forming the neighborhood $\mathcal{N}(i)$, where $M$ is set to 8 in our experiments. Please note that in the first layer, the neighborhood is defined based on spatial proximity, whereas in the subsequent layers, KNN is computed in the feature space, enabling the network to gather semantically or structurally similar points. Then, for every neighbor $p_j$ within the local neighborhood $\mathcal{N}(i)$, we compute the corresponding directional vector: $R_{ij} = p_j - p_i, j \in \mathcal{N}(i)$, and its local covariance matrix: $S_i = \frac{1}{M} \sum_{j \in \mathcal{N}(i)} R_{ij} R_{ij}^\top$. To build a rotation-invariant local coordinate frame for each point, we derive an $SO(3)$ orthonormal basis from the covariance matrix $S_i$. We approximate the principal direction $e_{1,i}$ using a single power-iteration step on $S_i$. Then, we use a fixed reference vector $t = [1, 0, 0]$ to generate the secondary direction $e_{2,i}$. If $t$ happens to be nearly collinear with $e_{1,i}$, we switch to an alternative reference vector $t = [0, 1, 0]$ to avoid degeneracy.

$$e_{1,i} = \frac{S_i v_0}{\|S_i v_0\|}, \qquad e_{2,i} = \frac{e_{1,i} \times t}{\|e_{1,i} \times t\|} \qquad (1)$$

where $v_0$ is a randomly sampled initialization vector. The third orthogonal direction is obtained as $e_{3,i} = e_{1,i} \times e_{2,i}$. The three orthogonal directions form the local rotation matrix: $E_i = [e_{1,i}, e_{2,i}, e_{3,i}] \in SO(3)$, which serves as a point-wise local coordinate frame for expressing directional information in a rotation-invariant manner. Then, we transform each directional vector into this frame as: $R_{ij}^{(\text{local})} = E_i^\top R_{ij}$. This projection maps directional vector into consistent orientation.

We apply convolution on both $R_{ij}^{(\text{local})}$ and $R_{ij}$, serving as two parallel branches. The former branch operates in the $SO(3)$ local frame, producing rotation-sensitive features $\mathcal{F}_i^{align}$, while the latter branch operates in the Euclidean frame, producing features $\mathcal{F}_i^{euclid}$ that preserve the original geometric structure. We then use a confidence weight $\alpha_i$ to adaptively fuse the two branches. To produce $\alpha_i$, we extract the trace of the covariance matrix $S_i$, characterizing the overall spatial extent of the neighborhood; and the anisotropy of $S_i$, describing how strongly the local geometry is oriented along specific directions.

$$tr_i = \text{tr}(S_i), \quad S_i^{\text{iso}} = \frac{tr_i}{3} I_3, \quad a_i = \|S_i - S_i^{\text{iso}}\|_F^2 \quad (2)$$

We then concatenate $tr_i$ and $a_i$, and pass them through a small CNN to get confidence weight $\alpha_i$. With $\alpha_i$, the two feature branches are adaptively fused, allowing them to complement each other: $\mathcal{F}_i = \alpha_i \mathcal{F}_i^{align} + (1 - \alpha_i) \mathcal{F}_i^{euclid}$. In addition, we incorporate two performance-enhancing modules, STE and ORL, proposed by HS-Pose(Zheng et al., 2023), which can better preserve translation information

and enhance outlier robustness. The features extracted by these modules are denoted as $\mathcal{F}_{STE}$ and $\mathcal{F}_{ORL}$, respectively. After that, we can get the final feature: $\mathcal{F}^{pc} = [\mathcal{F}_i, \mathcal{F}_{ORL}, \mathcal{F}_{STE}]$.

We insert our HyperS3 layer to 3D Graph Convolution Networks (3D-GCNs)(Lin et al., 2020), a well-known feature extraction network for 3D point cloud to construct our backbone. With our modified 3D-GCN backbone with HyperS3 layer, we obtain the point cloud feature for downstream process in our framework.

### 3.3. Quaternion Regression with Candidates to Final

#### 3.3.1. MOTIVATION AND CANDIDATES GENERATE

Most current works about 6D pose estimation focus on estimating rotation matrix(Di et al., 2022; Chen et al., 2024; Lin et al., 2022; 2021; Zhang et al.; Chen et al., 2021; Lin et al., 2023) by regressing two orthogonal columns of it and then recover the whole rotation matrix. For multiple-solution problem casued by symmetry, a natural method is firstly generating a group of candidates, then taking further aggregation or refinement. Because of the discontinuities in SO(3) manifold, regressing the whole rotation matrix as candidates is unreliable. Moreover, generating candidates for two separate orthogonal vectors is difficult to handle. Thus, we choose quaternion, as it provides a compact, singularity-free and not decoupled formulation, requiring only a single four-dimensional vector to describe a 6DoF rotation.

At first, we generate $K$ candidates $Q = \{q_1, \ldots, q_k\}$, where $K$ is set to 64 in our experiments. To this end, we first randomly sample $K$ noise vectors $\{z_1, \ldots, z_k\}$. Each noise vector is then concatenated with the point cloud feature $\mathcal{F}^{pc}$ to form the perturbed input feature $\mathcal{F}^{cand}_i = [\mathcal{F}^{pc}, z_i]$, which is used later to generate the $i$-th candidate. After constructing the perturbed input feature $\mathcal{F}^{cand}_i$ for all $K$ candidates, we reshape them together: $\mathcal{F}^{input} = [\mathcal{F}^{cand}_1, \ldots, \mathcal{F}^{cand}_K]$, so that each candidate can be generated independently and in parallel by a shared CNN once.

#### 3.3.2. CANDIDATES REFINEMENT.

At this stage, we refine the previously generated candidates. The simplest way to achieve this goal is leveraging a CNN to predict offset for each candidate on the tangent space of the unit quaternion hypersphere. Considering that the refinement corresponds to updating the rotation along a local direction in the tangent space, which can be treated as a first-order variation on the manifold. Therefore, performing updates in the quaternion tangent space provides a clear geometric interpretation and can be viewed as following a gradient-like direction on the manifold.

However, directly regressing without the candidates' in-

formation may lead to unreliable offset prediction. Thus, before estimating offset, we suggest a special candidates encoder to encode the candidates' information with $\mathcal{F}^{pc}$ for precise offset estimation. Our designs of candidates encoder are as follows. Given the set of candidates $Q$, we can first get their mean quaternion:

$$\tilde{q} = \frac{1}{K}\sum_{i=1}^{K} q_i, \quad q_{\mathrm{mean}} = \frac{1}{K}\sum_{i=1}^{K} \mathrm{sgn}\left(q_i^\top \tilde{q}\right) q_i. \quad (3)$$

Since $-q$ and $q$ represent the same physical rotation on $SO(3)$, we align the sign of each candidate according to the naive mean quaternion $\tilde{q}$. After that, we compute the residual between each candidate and the mean quaternion $q_{mean}$: $r_i = (q_{\mathrm{mean}})^{-1} \otimes q_i$. Then we map each residual quaternion $r_i = (w_i, x_i, y_i, z_i)$ from the hyperspherical manifold $S^3$ to its corresponding tangent space by converting it into an axis-angle representation $\delta_i \in \mathbb{R}^3$, which provides a linear and geometrically intuitive representation of the deviation in Euclidean space. Specifically, $\delta_i = \theta_i \mathbf{n}_i$, where $\theta_i$ denotes the rotation angle and $\mathbf{n}_i$ denotes the rotation axis. Then, we can compute the mean offset $\bar{\delta} = \frac{1}{K}\sum_{i=1}^{K} \delta_i$ from the mean rotation, describing their average deviation in tangent space. We further compute the averaged squared cosine similarity $\bar{\mu}$ in the quaternion space to quantify how tightly the candidates cluster around the mean rotation in the quaternion space $S^3$: $\bar{\mu} = \frac{1}{K}\sum_{i=1}^{K} \left|q_i^\top q_{mean}\right|^2$. To acquire the orientation information, we need to calculate the top three eigenvalues $\{\lambda_j\}_{j=1}^3$ and eigenvectors $\{v_j\}_{j=1}^3$ of the second moment matrix of offsets $M$. The algorithm is shown in 3.3.2.

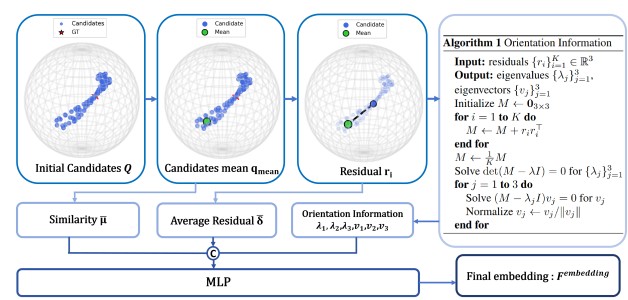

*Figure 2.* The illustration of the transformation process from the initial candidates $Q$ to the feature embedding $F^{embedding}$. The right demonstrates the extraction of orientation information. The left shows the extraction of other geometric information.

After computing all the descriptive quantities, we concatenate them into a single feature vector: $\mathcal{F}^{input} = \left[\bar{\delta}, \bar{\mu}, \lambda_1, \lambda_2, \lambda_3, v_1^\top, v_2^\top, v_3^\top\right]$, which serves as the input to the candidate encoder. Then, the encoder produces the $\mathcal{F}^{embedding}$, providing a rich and rotation-aware representation for subsequent stages. After passing through the encoder, the $\mathcal{F}^{embedding}$ and $\mathcal{F}^{pc}$ are fed into a linear layer to

obtain the fused representation $\mathcal{F}^{fused}$. With the candidates encoder's output, we finally apply a small CNN to obtain the predicted offsets: $\Delta Q = \{\Delta q_1, \ldots, \Delta q_k\}$ and the refined quaternion: $\Delta Q^{correct} = \{\Delta q_1 \otimes q_1, \ldots, \Delta q_k \otimes q_k\}$ We refine all the candidates along with their corresponding residual and then aggregate them by a linear layer to obtain the final quaternion: $q^{final} = W \cdot Q^{correct}$

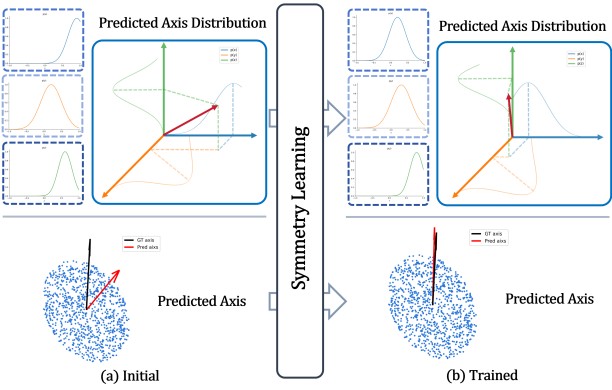

*Figure 3.* Process of the self-adaptive symmetry learning process. The figure depicts the evolution of the framework from an initial state (a) to a trained state (b). The top panels show the implicit probability distribution along $x, y, z$ axes.

## 3.4. Self-Adaptive Symmetry-Aware Design

### 3.4.1. SYMMETRY MOTIVATION AND OVERVIEW.

The multi-hypothesis nature of symmetric objects is caused by uncertainty in their symmetry axis or plane. Specifically, under a fixed viewpoint, rotations about a symmetry axis or reflections across a symmetry plane produce indistinguishable appearances, making multiple pose solutions feasible. Consequently, if the underlying symmetry axis or plane can be accurately identified, these equivalent poses can be explicitly parameterized, thus resolving the pose ambiguity. In this work, we discuss the two symmetries separately: rotational symmetry, and mirror symmetry.

In recent years, probabilistic methods have been adopted in several works(Yuan et al., 2025; Zhang et al., 2024c; Kolotouros et al., 2021) to address uncertainty in perception and vision tasks. Inspired by these, we formulate symmetry reasoning as distribution estimation problem over symmetry axes or planes by proposing a self-adaptive symmetry-aware network that only requires the type of symmetry rather than precise symmetry annotation. We can simply model a discrete mixture distribution $\pi_x, \pi_y, \pi_z$ over symmetry axes along the $x, y, z$-axis to quantify spatial uncertainty. However, for mirror symmetry, we face two challenges: the position of symmetry planes and the number of planes are both unknown. To handle them, we introduce point-cloud consistency measure that evaluates the predicted plane normals and suppresses the geometrically inconsistent planes.

### 3.4.2. ADAPTIVE NETWORK FOR SYMMETRY.

We convert the final quaternion $q^{final}$ into its axis-angle representation, denoted as $\mathcal{F}^{rot}$, which represents strong explicit rotational information. Then, both $\mathcal{F}^{rot}$ and $\mathcal{F}^{pc}$ are individually processed by CNN. After that, the two high-dimensional feature embeddings are concatenated and used to predict the probability of each axis.

**For rotational symmetry**, we can obtain the symmetry axis through a simple weighted combination:

$$n = \pi_x n_x + \pi_y n_y + \pi_z n_z, \tag{4}$$

where $n_x, n_y, n_z$ respectively denote the $x, y, z$-axis.

**For mirror symmetry**, the number of reflection planes is unknown. We assume that an object has three candidate symmetry-plane normals and evaluate each using a mirror-consistency metric based on the Chamfer distance, which measures how well the point cloud aligns with its reflection across a candidate plane. The resulting scores quantify geometric symmetry and are used to suppress outlier plane predictions. Specifically, we regard $n$ in Eq. 4 as the principal normal and additionally predict a secondary normal $n'$ which is enforced to be orthogonal to $n$. The third orthogonal direction $n''$ is obtained via their cross product. Symmetry scores are then computed for the three candidate plane normals $\{n, n', n''\}$. For each normal $u_j \in \{n, n', n''\}$, we obtain a reflected point cloud $p_i^{'(j)}$ by mirroring every point $p_i$ across the plane defined by $u_j$, where $p_c$ represents the centroid of the input point cloud:

$$\alpha_i^{(j)} = (p_i - p_c) \cdot u_j, \quad p_i^{'(j)} = p_i - 2\alpha_i^{(j)} u_j. \tag{5}$$

Given the original point set $P$ and its mirrored counterpart $P^{'(j)} = \{p_i^{'(j)}\}_{i=1}^N$, the one-sided Chamfer distance is:

$$d(P, P^{'(j)}) = \frac{1}{N} \sum_{a \in P} \min_{b \in P^{'(j)}} \|a - b\|, \tag{6}$$

where $N$ denotes the number of ponits. Similarly, we can define the reverse one-sided distance as $d(P^{'(j)}, P)$ in the same manner. The mirror-consistency score is computed by bidirectional Chamfer formulation:

$$\mathcal{L}_{geom}(P, u_j) = \frac{1}{2} \left( d(P, P^{'(j)}) + d(P^{'(j)}, P) \right). \tag{7}$$

Finally, the geometry-aware scores $s_j$ can be obtained:

$$s_j = \frac{\frac{1}{\mathcal{L}_{geom}(P, u_j) + \varepsilon}}{\sum_{j=1}^3 \frac{1}{\mathcal{L}_{geom}(P, u_j) + \varepsilon} + \varepsilon}, \tag{8}$$

where $\varepsilon$ is a small stabilizing constant. Given those $s_j$, we can suppress outlier planes.

*Table 1.* Results of GAParts Pose estimation in terms of per-part-class Rotation error and translation error. We use degree error (noted as °) and distance error (noted as cm) as metrics. Ln.=Line. F.=Fixed. Rd.=Round. Hg.=Hinge. Hl.=Handle. Sd.=Slider. Ld.=Lid. Bn.=Button. Dw.=Drawer. Dr.=Door. Kb.=Knob. Rot.=Rotation. Trans.=Translation.

| | | Method | Sd.Dw | Sd.Ld | Sd.Bn | Rd.F.HI | Hg.Kb | Ln.F.HI | Hg.HI | Hg.Dr | Hg.Ld | Avg |
|---|---|---|---|---|---|---|---|---|---|---|---|---|
| Seen | Rot.(°)↓ | GAPartNet | 5.03 | 7.82 | 2.41 | 12.74 | **4.74** | 10.39 | 10.15 | 8.46 | 7.72 | 7.71 |
| | | GASEM | 9.30 | 4.45 | 9.11 | 11.96 | 5.31 | 17.11 | 9.42 | 8.44 | 6.93 | 9.11 |
| | | GenPose++ | 1.95 | 19.46 | 1.97 | 29.94 | 44.23 | 20.94 | 9.76 | 9.83 | 4.62 | 15.86 |
| | | RFMPose | 1.53 | 21.11 | 1.87 | 31.94 | 57.97 | 15.13 | 8.05 | 11.45 | 4.20 | 17.03 |
| | | DFGAP | 1.79 | 3.58 | 2.44 | 5.12 | 9.03 | 7.71 | **6.04** | 5.63 | 8.21 | 5.51 |
| | | SAFAG(Ours) | **1.44** | **0.46** | **0.69** | **3.56** | 8.72 | **3.07** | 7.77 | **1.00** | **2.39** | **3.23** |
| | Trans.(cm)↓ | GAPartNet | 0.075 | 0.032 | 0.009 | 0.078 | 0.010 | 0.015 | 0.039 | 0.038 | 0.038 | 0.037 |
| | | GASEM | 0.069 | 0.023 | 0.005 | 0.051 | 0.005 | 0.015 | **0.031** | 0.047 | 0.080 | 0.036 |
| | | GenPose++ | **0.024** | 0.039 | 0.014 | 0.018 | 0.021 | 0.072 | 0.072 | 0.048 | 0.029 | 0.035 |
| | | RFMPose | 0.046 | 0.049 | 0.031 | 0.035 | 0.069 | 0.103 | 0.087 | 0.079 | 0.042 | 0.060 |
| | | DFGAP | 0.037 | 0.014 | 0.008 | 0.037 | 0.005 | 0.011 | 0.024 | 0.025 | 0.019 | 0.020 |
| | | SAFAG(Ours) | 0.032 | **0.012** | **0.002** | **0.004** | **0.003** | **0.010** | 0.056 | **0.014** | **0.018** | **0.016** |
| Unseen | Rot.(°)↓ | GAPartNet | 14.62 | 29.17 | 9.21 | 19.89 | 16.89 | 36.54 | 64.31 | 38.57 | 19.18 | 27.59 |
| | | GASEM | 20.72 | 24.04 | 8.28 | 27.41 | **12.86** | 33.85 | 57.66 | 22.62 | 57.66 | 29.45 |
| | | GenPose++ | 14.73 | 26.51 | 39.90 | 10.38 | 86.45 | 16.61 | 18.11 | 16.00 | 56.26 | 31.66 |
| | | RFMPose | 6.33 | 26.81 | 53.22 | 8.20 | 76.94 | 16.10 | 28.25 | 17.85 | 66.78 | 33.39 |
| | | DFGAP | 4.15 | 10.28 | 5.98 | 4.33 | 19.17 | 20.03 | **16.69** | 12.87 | 13.24 | 11.86 |
| | | SAFAG(Ours) | **3.74** | **0.41** | **3.12** | **3.94** | 28.98 | **15.83** | 33.35 | **3.40** | **4.71** | **10.83** |
| | Trans.(cm)↓ | GAPartNet | 0.318 | 0.076 | 0.042 | 0.091 | 0.038 | 0.164 | 0.539 | 0.131 | 0.415 | 0.201 |
| | | GASEM | 0.165 | 0.385 | 0.014 | 0.052 | 0.019 | 0.226 | 0.261 | 0.087 | 0.345 | 0.172 |
| | | GenPose++ | 0.133 | 0.024 | 0.037 | 0.021 | 0.255 | 0.033 | **0.079** | 0.054 | 0.094 | 0.081 |
| | | RFMPose | 0.133 | 0.049 | 0.062 | 0.051 | 0.129 | 0.113 | 0.123 | 0.081 | 0.202 | 0.105 |
| | | DFGAP | 0.063 | 0.035 | 0.012 | 0.053 | **0.076** | 0.126 | 0.163 | 0.033 | 0.192 | 0.084 |
| | | SAFAG(Ours) | **0.058** | **0.012** | **0.004** | **0.004** | 0.025 | **0.022** | 0.399 | **0.028** | **0.046** | **0.066** |

## 3.5. Training Strategy and Loss function.

For categories of all kind of symmetry, we divide the training proceeding into two stages.

**For rotational symmetry**, in the warm-up stage, we only constrain the candidate quaternions and their distribution using symmetry-aware loss functions without the predicted symmetry axis. This encourages the candidates to form a relatively correct distribution around corresponding equivalent solutions.

We first generate symmetry-equivalent solutions based on the ground-truth quaternion $q_{gt}$. We sample a range of rotation angles around each axis to obtain all rotation-equivalent solutions $Q_k^{eq} = \{q_{i,k}^{eq}\}_{i=1}^{N_{eq}}, \quad k \in \{x, y, z\}$. $N_{eq}$ denotes the number of sampled equivalent poses, and we set $N_{eq}$ as 36. Then, for every predicted candidate $q_j^{cand}$, we calculate its minimum angular distance to all equivalent quaternions in $Q_k^{eq}$. Then, we average this minimum distance over all $K$ candidates and select the axis that gives the smallest average discrepancy:

$$\mathcal{L}_{\text{warm}}^{\text{rot}} = \min_{k \in \{x,y,z\}} \left( \frac{1}{K} \sum_{j=1}^{K} \min_{1 \leq i \leq N_{\text{eq}}} 2 \arccos\left( \left| \langle q_j^{\text{cand}}, q_{i,k}^{\text{eq}} \rangle \right| \right) \right). \quad (9)$$

Specifically, we replace the original min operation with a temperature-controlled weighted average with $\beta = 10$, which provides a smoother approximation and helps improve training stability. The same soft relaxation strategy is consistently adopted in the following formulations.

After the warm-up stage, we replace the hypothetical symmetry axes with the predicted symmetry axis to generate the corresponding symmetry-equivalent solutions $Q^{eq} = \{q_i^{eq}\}_{i=1}^{N_{eq}}$. For candidates, we follow the same procedure as in the warm-up stage, except that equivalent solutions are generated using the predicted axis:

$$\mathcal{L}_{cand}^{rot} = \frac{1}{K} \sum_{j=1}^{K} \min_{1 \leq i \leq N_{\text{eq}}} 2 \arccos\left( \left| \langle q_j^{cand}, q_i^{eq} \rangle \right| \right) \quad (10)$$

And for the final quaternion $q_{final}$, we compute its loss as the minimum angular deviation from the equivalent set:

$$\mathcal{L}_{final}^{rot} = \min_{1 \leq i \leq N_{\text{eq}}} 2 \arccos\left( \left| \langle q_{final}, q_i^{eq} \rangle \right| \right) \quad (11)$$

We observe that directly supervising the angular difference around the symmetry axis is considerably less effective, as shown in our ablation studies. In contrast, explicitly generating symmetry-equivalent solutions provides a richer supervisory signal by implicitly encouraging the network to learn the distribution of equivalent rotations.

**For mirror symmetry**, we define the loss function as:

$$\mathcal{L}^{mirror} = \mathcal{L}_{angle}^{mirror} + \mathcal{L}_{geom}^{mirror}, \quad (12)$$

where $\mathcal{L}_{angle}^{mirror}$ follows exactly the same procedure as the rotational-symmetry loss in both two stages, except that the symmetry-equivalent solutions are generated through reflection rather than rotation.

*Table 2.* Result of backbone analysis on GAParts pose estimation. We use the averaged rotation error (noted as °) and the averaged percentage (noted as %) of predictions within 10°10 cm, 5°5 cm, and 5°2 cm thresholds of all part of GAParts as metrics.

| Backbone | Seen | | | | Unseen | | | |
|---|---|---|---|---|---|---|---|---|
| | Avg.Rot.(°)↓ | 10°10cm↑ | 5°5cm↑ | 5°2cm↑ | Avg.Rot.(°)↓ | 10°10cm↑ | 5°5cm↑ | 5°2cm↑ |
| HS-Layer(Zheng et al., 2023) | 4.50 | 88.43 | 75.00 | 63.48 | 14.15 | 67.46 | 49.62 | 36.68 |
| Ours | **3.23** | **91.84** | **81.48** | **69.91** | **10.83** | **70.74** | **55.77** | **44.26** |

For the mirror-consistency loss term $\mathcal{L}_{geom}^{mirror}$, during the warm-up stage, we evaluate it over the three canonical plane normals $u_j \in \{x, y, z\}$ and average the results over all candidates. In the main stage, the canonical axes are replaced by the predicted mirror-plane normals $\{u_j\}_{j=1}^{\Omega}$:

$$\mathcal{L}_{geom}^{mirror} = \frac{1}{\Omega} \sum_{j=1}^{\Omega} \mathcal{L}_{geom}(P, u_j), \qquad (13)$$

where $\Omega$ denotes the number of symmetry planes predicted in the main stage, and $\Omega = 3$ in the warm-up stage.

**For asymmetric objects**, we directly measure the angular deviation between the prediction and the ground-truth pose.

# 4. EXPERIMENTS

## 4.1. Experiment Setup

**Data Preparation** We implement our framework on the GAPartNet dataset (Geng et al., 2023), which contains 9 GAPart classes (such as lids, handles, etc.) across 27 object categories. The dataset provides part-level annotations for 8,489 instances of parts from 1,166 objects, collected from PartNet-Mobility(Xiang et al., 2020) and AKB-48(Liu et al., 2022).

**Evaluation Metrics** We evaluate the rotation and translation estimation performance using the metrics of the rotation error (noted as °) and translation error (noted as cm). Moreover, to directly compare errors in rotation and translation, we also adopt Average Precision (AP) under 5°2cm, 5°5cm,10°5cm, 10°10cm error as our evaluation metric.

**Implementation Details** The input point clouds are sampled into 1,024 points. We employ the Adam optimizer. Training and validation batch sizes both are 16. All the experiments are implemented on four NVIDIA GeForce RTX 4090 GPUs with 24GB memory.

## 4.2. Pose Estimation Results and Discussion

We report the part pose estimation evaluated on the GAPartNet dataset. The experimental results are shown in Table. 1. We select five comparison methods from recent top conferences, three of them are particularly designed for GAParts, GAPartNet(Geng et al., 2023), GASEM(Liu et al., 2025) and DFGAP(Yuan et al., 2025). Two of them are designed for category-level object, GenPose++(Zhang et al., 2024a)

*Table 3.* Results of explicit equivalent solution analysis on GAParts pose estimation.We report the rotation error (noted as °) of Rotational-Symmetry objects. E.S. denotes the explicit equivalent solution, while S.E. denotes the Symmetry axis error.

| | Seen(°)↓ | | Unseen(°)↓ | |
|---|---|---|---|---|
| | E.S. | S.E | E.S. | S.E |
| Sd.Ld | **0.46** | 22.23 | **0.41** | 28.45 |
| Sd.Bn | **0.69** | 5.11 | **3.12** | 6.54 |
| Rd.F.HI | **3.56** | 12.38 | **3.94** | 4.99 |
| Hg.Kb | **8.72** | 29.38 | **28.98** | 48.41 |

and RFMPose(Ouyang et al., 2026). For fair comparison, we unify the input as segmented gapart point clouds or RGB image, and train them on GAPartNet dataset. Compared with existing methods, our approach achieves the best performance in the vast majority of GAParts in the GAPartNet dataset. For seen categories, in terms of rotation estimation, our method achieves rotation errors within **10°** across all categories. The average rotation error (3.23°) is reduced by **39.1%** compared to the second-best method. For unseen categories, the average rotation error (10.83°) is reduced by **8.68%** compared to the second-best method. For translation, our method achieves the lowest errors, corresponding to approximately **20%** and **19%** reductions compared to the second-best methods, respectively. Specifically, our method outperforms the main baseline, GAPartNet and GASEM, in almost all items. Since they still tackle the pose estimation by the paradigm of NPCS, the pose estimation result heavily relies on the precise NPCS prediction. Compared with the current SOTA method, DFGAP, our method still achieves highly competitive performance. DFGAP is a depth-free method which eliminates the need for depth input. Although DFGAP has minimized the impact of lacking depth and adopted a loss function specifically designed for symmetry, its single-modality input still inevitably affects the final performance. The comparison result with two category-level methods GenPose++ and RFMPose proves the necessity of our method. They adopt the most advanced generative model. However, ignoring the richer symmetry of components relative to the object and the lack of special design for symmetry lead to poor performance on the GAPartNet dataset, especially on the categories with high symmetry, such as Sd.Bn, Hg.Kb and Hg.Ld.

## 4.3. Ablation Studies

**Symmetry Analysis for Rotation Estimation.** We evaluate the effectiveness of symmetry-aware network in rotation estimation. As shown in Table. 4, For seen categories, the

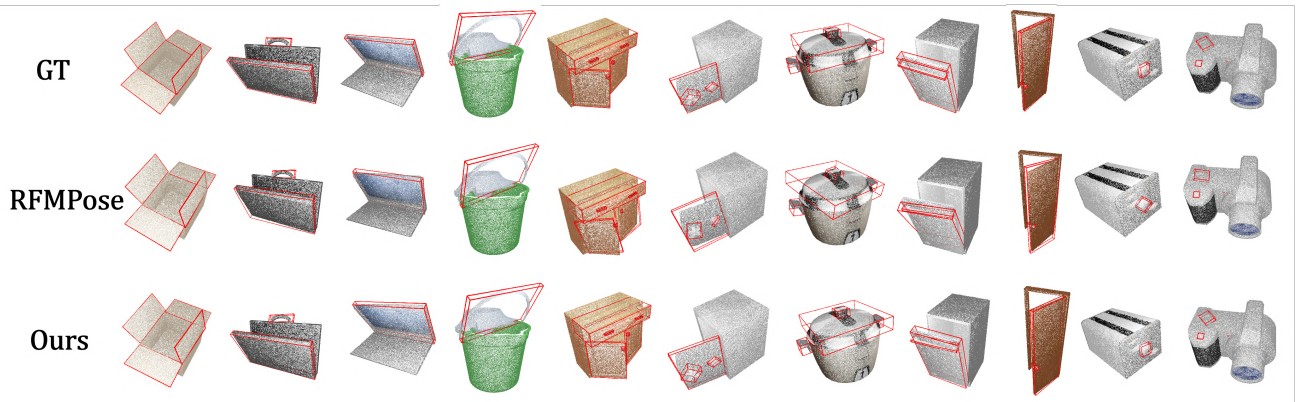

*Figure 4.* Qualitative results on GAParts pose estimation. We compare our method with RFMPose(Ouyang et al., 2026). More qualitative results can be found in supplementary materials.

*Table 4.* Results of symmetry analysis on GAParts pose estimation for part classes with rotational or mirror symmetry in terms of Rotation error. We use degree error (noted as °) as metrics.

|  | Sym-Aware | **Rotational** | **Mirror** | **Avg** |
|---|---|---|---|---|
| **Seen(°)↓** | ✗ | 25.01 | 10.05 | 17.53 |
|  | ✓ | **3.35** | **3.55** | **3.45** |
| **Unseen(°)↓** | ✗ | 28.90 | 56.27 | 42.59 |
|  | ✓ | **9.11** | **14.32** | **11.72** |

*Table 5.* Sensitivity analysis of parameters $N_{eq}$ and $K$ on GAParts pose estimation in terms of rotation error. Degree error (°) is used as the evaluation metric. The default setting is shown in bold.

| $K$ | **Seen (°)** | **Unseen (°)** | **Inference FPS** |
|---|---|---|---|
| 32 | 3.70 | 4.68 | 74.14 |
| **64** | 3.56 | 3.94 | 74.19 |
| 128 | 3.23 | 3.99 | 73.07 |

| $N_{eq}$ | **Seen (°)** | **Unseen (°)** | **Training FPS** |
|---|---|---|---|
| 18 | 5.97 | 11.77 | 6.84 |
| **36** | 3.56 | 3.94 | 4.33 |
| 72 | 3.27 | 3.55 | 2.55 |

average error is decreased by **79.9%** and for unseen categories, the average error achieves a **67.3%** reduction.

**Backbone Analysis.** We compare our proposed backbone with the HS-Layer on GAParts pose estimation in Table. 2. In terms of average rotation error, our method reduces the error by **28.2%** and **7.8%** respectively for seen and unseen categories. Meanwhile, our method consistently improves all threshold-based metrics, yielding gains of **3.41%**, **9.48%**, and **6.51%** on seen categories, and **3.28%**, **6.15%**, and **7.58%** on unseen categories under the 10°10 cm, 5°5 cm, and 5°2 cm thresholds, respectively.

**Explicit Equivalent Solution Analysis.** We report the rotation error obtained using the explicit equivalent solution loss and the symmetry-axis error loss (supervising the angular difference around the symmetry axis) on rotational-symmetry objects. As can be observed in Table. 3, optimiz-

*Table 6.* Robustness analysis of our method to symmetry-type mislabeling on rotationally symmetric objects. Degree error (°) is used as the evaluation metric.

| **Setting** | **Seen (°)** | **Unseen (°)** |
|---|---|---|
| Correct rotational symmetry | 3.35 | 9.11 |
| Mistaken as mirror symmetry | 10.06 | 14.48 |
| Mistaken as asymmetric | 25.01 | 28.90 |

ing with the explicit equivalent solution loss leads to lower rotation errors. In particular, the rotation error is reduced by approximately **60-80%** for the seen categories, and by around **20-60%** for the unseen categories.

**Parameter Sensitivity Analysis.** We further conduct experiments with different values of $N$ and $K$ to evaluate the sensitivity of our model to these parameters, as shown in Table 5. For K, the model is not particularly sensitive to its value. Within a reasonable range, varying K leads to only minor changes in rotation performance, and K has a negligible impact on the computational time. This is because increasing K only expands the candidate dimension in the lightweight prediction head, while the computationally dominant backbone remains unchanged. For $N_{eq}$, the model shows some sensitivity. A larger $N_{eq}$ provides denser supervision over the equivalent solution set and improves optimization performance, but it also increases the computational cost. This is because more sampled equivalent poses are involved in the loss computation during training. Therefore, we choose a moderate value of $N_{eq}$ to balance training efficiency and performance.

**Impact of Symmetry Misclassification** To evaluate the robustness of our method, we conduct additional experiments on rotationally symmetric objects by intentionally mislabeling their symmetry types as either mirror-symmetric or asymmetric, as shown in the Table 6. The results show that when a rotationally symmetric object is treated as mirror-symmetric, the model can still capture part of the underlying symmetry and equivalent-set structure. In contrast, when

it is treated as asymmetric, the rotational performance degrades noticeably, because the model cannot recognize the underlying symmetry and will incorrectly penalize pose predictions that are actually valid equivalent solutions.

### 4.4. Real-world Evaluation

We further evaluate the generalizability of our method in real-world scenarios. We select four GAParts classes for evaluation: Hinge Door, Hinge Lid, Slider Drawer, and Hinge Handle which can be captured from storage box, laptop, bucket and drawer, respectively. Simply put, we first construct the fused TSDF by multiple partial point cloud from consecutive video frames. Then, with complete mesh, we can obtain the pose annotation of corresponding GAPart. While evaluation, we input the segmented point cloud to our trained model. Qualitative results are shown in Fig. 4.4. The results demonstrate the robustness and generalizability of our framework in the real world.

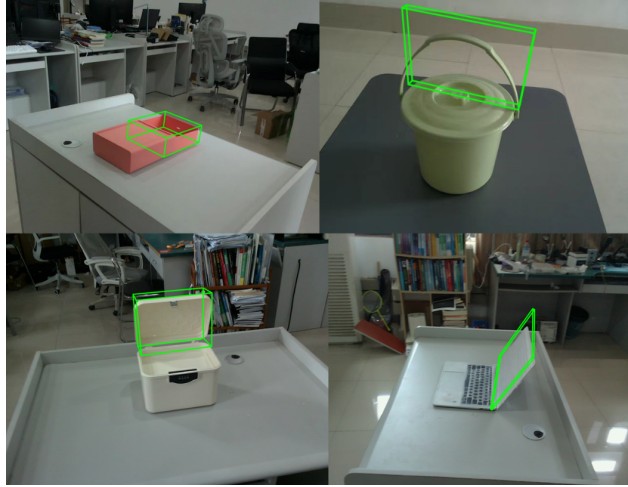

*Figure 5.* Qualitative results on GAParts pose estimation in real-world. We evaluate the effectiveness of our method in real-world. The dataset includes Slider Drawer (top-left), Hinge Handle (top-right), Hinge Door (bottom-left), and Hinge Lid (bottom-right).

## 5. CONCLUSION

In this paper, we propose a symmetry-annotation-free method for precise and robust GAParts pose estimation. By introducing a two-stage refinement framework and a specific symmetry learning strategy, our method effectively addresses the multi-hypothesis ambiguity induced by object symmetries without requiring explicit symmetry annotations. Extensive experiments on the GAPartNet dataset demonstrate that our approach achieves superior performance compared to various latest methods. Our framework is well suited for real-world robotic perception and can be readily applied to embodied AI tasks, such as robot manipulation, augmented reality, and 3D scene understanding.

## ACKNOWLEDGEMENTS

This work is supported in part by National Natural Science Foundation of China under Grant 62302143, Natural Science Foundation of China 62272144, and the Anhui Provincial Natural Science Foundation 2408085J040.

## Impact Statement

This paper presents work whose goal is to advance the field of Machine Learning. There are many potential societal consequences of our work, none which we feel must be specifically highlighted here.

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

# A. Dataset Extended

## A.1. Introduction of GAParts

GAPartNet(Geng et al., 2023) is a large-scale interactive part-centric dataset that contains 1,166 articulated objects from the PartNet-Mobility(Xiang et al., 2020) dataset and the AKB-48(Liu et al., 2022) dataset. Xiang et al. propose PartNet-Mobility, a large-scale 3D interactive model dataset comprising 2,346 articulated object models from 46 common indoor categories, with over 14,000 movable parts annotated with rich kinematic motions and dynamic interactive attributes. Articulated objects interaction tasks can be seen in Fig. A.1. GAPartNet also relies on AKB-48, a largescale real-world dataset comprising 2,037 3D articulated object models in 48 categories, each enriched with detailed annotations. In GAPartNet, GAPart represents a set of object parts that are visually generalizable and consistently actionable, bridging the gap between perception and robotic manipulation. As illustrated in Fig. A.1, GAPartNet identifies 9 common GAPart classes across 27 object categories: line-fixed handle, round-fixed handle, hinge handle, hinge lid, slider lid, slider button, slider drawer, hinge door and hinge knob.

GAParts are designed for scenarios where robots interact with functionally consistent and reusable manipulation units that can be shared across different object categories. Under this definition, GAParts may correspond to either large-scale components (e.g., doors and drawers) or small-scale components (e.g., buttons and knobs), as long as these parts serve as reusable manipulation units with consistent interaction patterns. Accordingly, whether the robot operates on the object as a whole or on a local part depends on the task-relevant functional unit. The key advantage is that GAParts provide a unified representation for cross-category functional parts, enabling the model to generalize to unseen objects by leveraging shared functional structure.

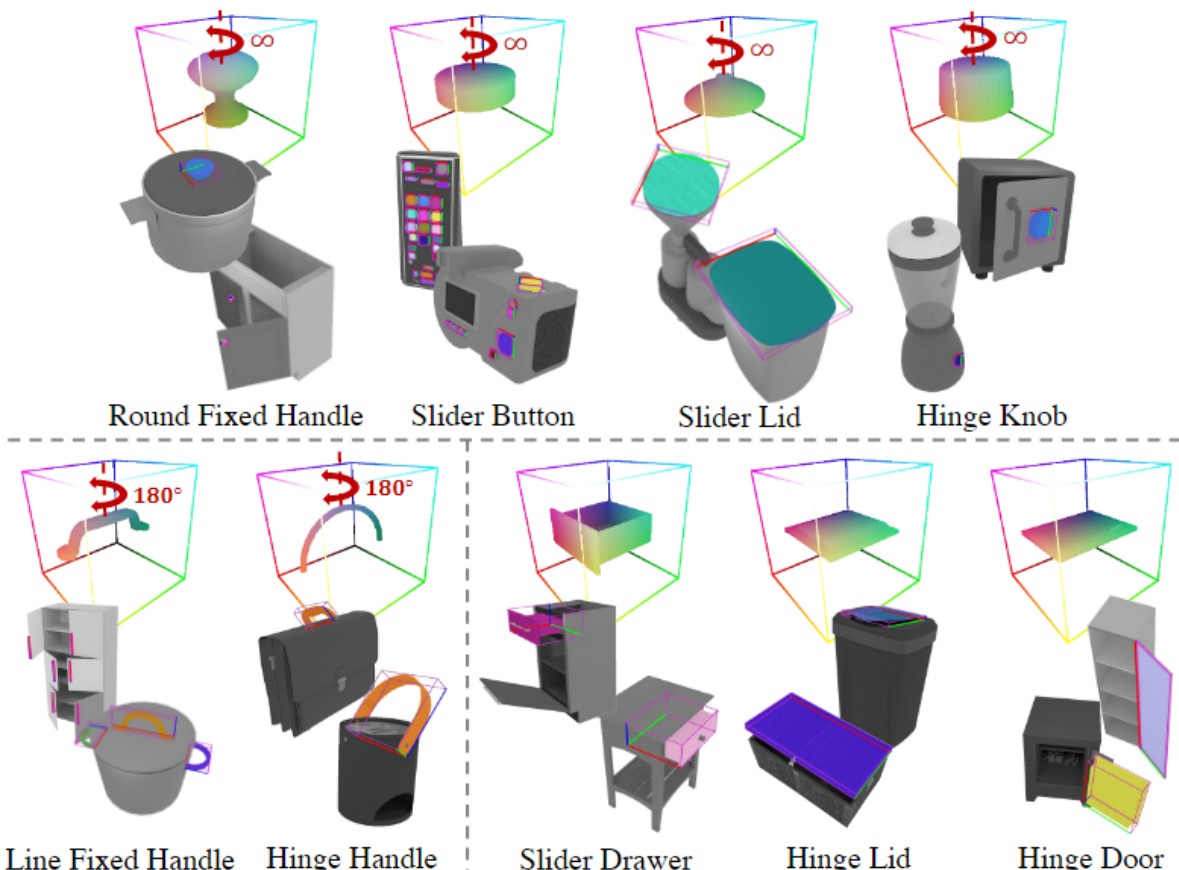

*Figure 6.* GAPart definition. Figure adapted from GAPartNet.

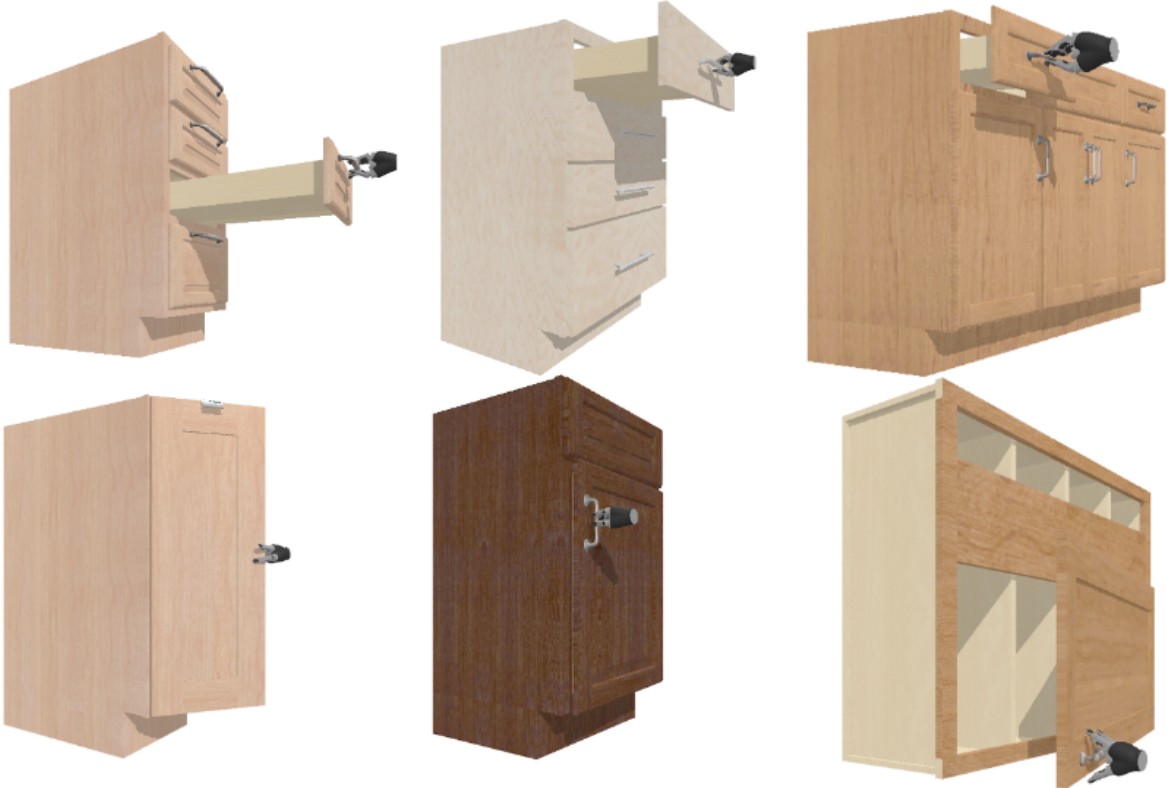

*Figure 7.* Articulated Object Interaction tasks. Figure adapted from PartNet-Mobility.

## B. Experiment Extended

### B.1. GAParts Manipulation

We also present the qualitative manipulation results achieved by our framework, along with comprehensive manipulation evaluations conducted in real-world environments. We set four different kinds of tasks, pulling drawer open, toggling power-strip switch, lifting a lid off a bucket and pulling appliance door open. We use xArm and its gripper to validate the robustness and precision of our framework in real-world interaction experiments. Qualitative results can be observed in Fig.B.1

We use the xArm robotic arm for our part-based object manipulation experiments due to its high precision, flexibility, and user-friendly design. As a state-of-the-art collaborative robot by UFACTORY, xArm's multiple degrees of freedom and seamless integration with ROS and Python-based SDKs enable it to handle complex tasks such as grasping, assembling, and precise object manipulation. Additionally, UFACTORY's comprehensive documentation and active community support further enhance its adaptability, making it an excellent choice for advancing our research in robotic perception and interaction.

### B.2. First-stage Candidate Quality Analysis

To better understand the quality of the first-stage generated candidates, we further conduct a quantitative analysis on several selected categories from GAPart. Specifically, we evaluate the candidates from two perspectives: 1) the rotation error between the generated candidates and the ground-truth pose, denoted as *Cand vs. GT*; and 2) the rotation error between the generated candidates and the final predicted pose, denoted as *Cand vs. Final*. For reference, we also report the rotation error between the final prediction and the ground truth, denoted as *GT vs. Final*.

As shown in Table. 7, the results show that the generated candidates, the final prediction, and the ground-truth pose are all relatively close in the pose space, since Cand vs GT and Cand vs Final are highly consistent with only small differences. This indicates that the first-stage candidates already form a reasonable pose neighborhood. Moreover, the error of GT vs

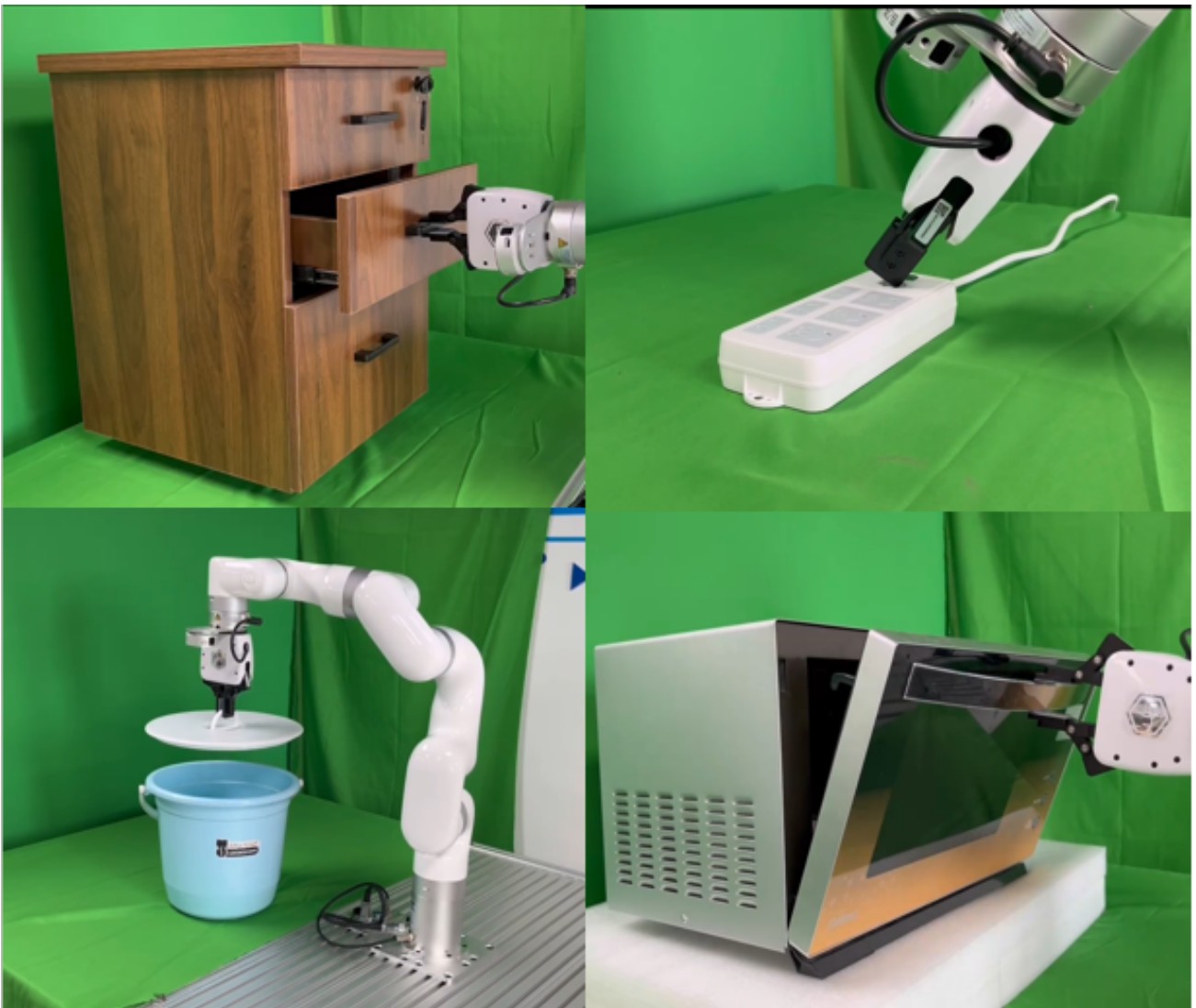

*Figure 8.* Qualitative results of part-based object manipulation in the real world.

Final is further reduced, showing that aggregating the candidates can move the final prediction closer to the ground truth. However, since the final prediction is produced by aggregating the whole candidate set rather than selecting the oracle-best candidate, it is still affected by candidates with larger deviations and therefore cannot perfectly match the ground truth. Overall, these results verify that the candidate set is informative and that the second stage can further improve the prediction quality.

### B.3. Generalization on other dataset

To further evaluate the cross-dataset generalization ability of our method, we conduct additional experiments on the ClearPose dataset. Specifically, we select four representative object categories, including plate, water cup, wine cup, and bowl. These categories exhibit clear symmetry properties, making them suitable for evaluating the generalization of our symmetry-aware framework. For a fair comparison, we use the ground-truth depth provided by ClearPose to generate point clouds as input to our model.

As shown in Table.8, these results show that our method remains effective on this non-part-centric dataset, suggesting that our framework has promising generalization ability beyond the GAPart Dataset.

*Table 7.* First-stage candidate quality analysis on selected GAPart categories. Degree error (°) is used as the evaluation metric. *Cand vs. GT* denotes the error between the generated candidates and the ground-truth pose, *Cand vs. Final* denotes the error between the generated candidates and the final prediction, and *GT vs. Final* denotes the error between the final prediction and the ground truth.

| Category | Setting | Cand vs. GT | Cand vs. Final | GT vs. Final |
|----------|---------|-------------|----------------|--------------|
| Rd.F.HI | Seen | 8.04 | 8.30 | 3.56 |
| Rd.F.HI | Unseen | 8.60 | 8.79 | 3.94 |
| Sd.Ld | Seen | 5.51 | 5.55 | 0.46 |
| Sd.Ld | Unseen | 5.74 | 5.84 | 0.41 |
| Sd.Bn | Seen | 6.78 | 6.85 | 0.69 |
| Sd.Bn | Unseen | 7.53 | 7.22 | 3.12 |
| Hg.Dr | Seen | 15.39 | 15.08 | 1.00 |
| Hg.Dr | Unseen | 15.73 | 15.70 | 3.40 |
| Hg.Ld | Seen | 16.37 | 16.37 | 2.39 |
| Hg.Ld | Unseen | 16.87 | 16.95 | 4.71 |

*Table 8.* Cross-dataset generalization results on selected categories from the ClearPose dataset. Degree error (°) and distance error (cm) are used as evaluation metrics.

| Metric | Plate | Water Cup | Wine Cup | Bowl |
|--------|-------|-----------|----------|------|
| Rot. (°) | 2.42 | 1.79 | 1.52 | 3.07 |
| Trans. (cm) | 0.008 | 0.007 | 0.007 | 0.008 |

## B.4. Details of Ablation Studies

In this section, we further explain some settings and implement details of our ablation studies mentioned in main paper. Specifically, the analysis is organized into three primary components: 1) the effectiveness of our self-adaptive symmetry-aware design for rotation estimation; 2) the performance gains attributed to HyperS3; and 3) the comparison between the explicit equivalent solution and symmetry axis error solution for loss function.

**Symmetry Analysis for Rotation Estimation.** We report the effectiveness of our self-adaptive symmetry-aware design. Specifically, we conduct this ablation study by removing this module. The model is trained using only the ground-truth pose for loss computation, rather than accounting for equivalent solutions generated by object symmetries obtained by this module. We conduct the analysis on GAParts pose estimation for part classes with rotational and mirror symmetry.

**Backbone Analysis.** To evaluate the impact of our HyperS3 layer, we conduct a comparative study by replacing it with the HS-layer. We then assess the performance between these two backbones on the GAParts dataset. By using two complementary metrics: rotation error and the average percentage of correct predictions (within a specified error threshold), we can better present the performance gap.

**Explicit Equivalent Solution Analysis.** To explain the reason for generating explicit equivalent solutions for loss computation, we compare our approach against a baseline that directly supervises the angular difference around the symmetry axis of rotational classes in GAParts. Specifically, we constrain pose discrepancy by comparing the predicted symmetry axis directions under the predicted and ground-truth poses. This suggests that the module effectively learns the underlying distribution of equivalent rotations.

**Parameter Sensitivity Analysis.** We further study the influence of two key hyperparameters, namely the number of rotation candidates $K$ and the number of sampled equivalent poses $N_{eq}$. For the candidate number $K$, we vary its value while keeping other settings unchanged. For $N_{eq}$, we sample different numbers of equivalent poses from the symmetry-induced solution set during training.

**Impact of Symmetry Misclassification.** To further evaluate the robustness of our method to inaccurate symmetry annotations, we intentionally assign incorrect symmetry types to rotationally symmetric objects. Specifically, we replace the correct rotational symmetry annotation with either mirror symmetry or no symmetry, then train and evaluate the model under these modified settings.

# C. Details of Real-world Evaluation

In this section, we provide a detailed description of the process of the experiment on real-world evaluation. The experimental procedure is organized into three primary stages: 1) Data preparation; 2) Data processing and ground-truth generation; and 3) Model inference and qualitative analysis.

## C.1. Data Preparation

We perform a 360° surrounding scan with varying elevations for each target object (drawer, bucket, safe, and laptop) using an Intel RealSense L515 depth camera to ensure full geometric coverage.The technical details are as follows:

We utilize a RealSense L515 camera to capture synchronized high-resolution RGB-D sequences. The color stream is configured at a resolution of $1280 \times 720$ pixels in BGR8 format, while the depth stream is captured at $1024 \times 768$ pixels in Z16 format. This high-density depth information ensures the geometric fidelity of the small part classes in our study. To ensure exposure stability, the camera pipeline is warmed up for several frames before recording. The acquisition frequency is set to 4 FPS, which maintains a sufficient frame-to-frame overlap for subsequent point cloud registration while avoiding excessive data redundancy. For each object, we collect approximately 120 RGB-D image pairs, providing a dense multi-view representation that maintains spatial continuity throughout the 360-degree scan. Each frame pair is stored in lossless PNG format and the depth values are preserved in millimeters to facilitate accurate 6D pose estimation.

## C.2. Data Processing and Ground-truth Generation

Following data acquisition, we implement a multi-stage preprocessing and integration pipeline to convert raw RGB-D sequences into high-fidelity 3D mesh models and recover optimized camera trajectories. This procedure performs depth-to-point cloud conversion, geometric filtering, ICP-based odometry, and volumetric TSDF fusion.

### C.2.1. VOLUMETRIC RECONSTRUCTION AND ODOMETRY

Each depth frame is back-projected into a 3D point cloud using calibrated camera intrinsics $\mathbf{K}$:

$$\mathbf{K} = \begin{bmatrix} 730.75 & 0 & 484.80 \\ 0 & 732.00 & 397.42 \\ 0 & 0 & 1 \end{bmatrix}, \tag{14}$$

with a depth scaling factor of $2.5 \times 10^{-4}$m per unit. Depth values are truncated to the range $[0.15, 1.20]$m to isolate the near-field region containing the target object. These parameters are determined by the intrinsic calibration and quantization characteristics of the specific sensor.

A pass-through filter restricts points to $x, y \in [-0.6, 0.6]$m in the camera coordinate system. Large planar structures (*e.g.*, tabletops) are removed via RANSAC-based plane segmentation with an $8\,\text{mm}$ distance threshold and 800 iterations. Camera extrinsics are estimated incrementally via frame-to-frame Point-to-Plane ICP odometry. We employ a two-stage alignment with maximum correspondence distances of 3cm and 1.5cm, utilizing a robust Tukey loss ($k = 0.02$). To ensure the fidelity of the reconstructed model, only frames meeting the stringent registration criteria—specifically a fitness score exceeding 0.20 and an inlier RMSE below 3cm are integrated into the Scalable TSDF Volume. This filtering process prevents tracking inaccuracies from degrading the final surface mesh, which is subsequently extracted and refined through vertex normal computation.

### C.2.2. GLOBAL OBJECT LOCALIZATION AND REFINEMENT

Based on the reconstructed mesh and optimized camera extrinsics, we compute the transformation from the world coordinate system to the object-centric coordinate system.

- **Initial Localization:** Depth frames are accumulated in world space using estimated poses to provide a stable observation of the scene. We estimate the tabletop plane via RANSAC to define the global vertical direction and retain points within a height range of $[0.01, 0.35]$m above this plane. The largest connected cluster within this filtered region is extracted as the target object.

- **Registration of Scanned Mesh:** To estimate the object pose $\mathbf{T}_{wo}$, a pre-scanned reference mesh is sampled into a point cloud. A coarse alignment is achieved through FPFH feature matching, followed by a discrete yaw search around the vertical axis to resolve rotational ambiguities.

- **Ground-truth Refinement:** To further ensure precision, the fused TSDF surface serves as a reference for geometric calibration. We manually select corresponding points on the fused surface and the scanned reference mesh to obtain a reliable initialization via SVD-based closed-form alignment. This is followed by fine-scale Point-to-Plane ICP to correct residual errors in $\mathbf{T}_{wo}$.

### C.2.3. Part-level Registration and Extraction

The rigid transformation between each object part and the full object is determined through a mesh-level registration procedure.

- **Pre-defined Part Alignment:** When a scanned part mesh is available, it is aligned to the full scanned reference mesh. Initial rigid transformations are estimated via SVD-based alignment from manually selected surface points, followed by multi-stage Point-to-Plane ICP refinement to define the fixed part-to-object transformation $\mathbf{T}_{po}$.

- **Interactive Part Extraction:** For objects without pre-defined part meshes, we interactively extract parts from the full scanned mesh. We select surface points to fit a local plane and define a 2D bounding region. Triangles whose centroids lie within these bounds and satisfy a normal tolerance are retained as the part mesh.

### C.2.4. 6D Pose Composition

Finally, we compute the part-to-camera transformation $\mathbf{T}_{pc}^{(i)}$ for each frame $i$ by composing the estimated transformations:

$$\mathbf{T}_{pc}^{(i)} = \left(\mathbf{T}_{wc}^{(i)}\right)^{-1} \cdot \mathbf{T}_{wo} \cdot \mathbf{T}_{po}, \tag{15}$$

where $\mathbf{T}_{wc}^{(i)}$ is the optimized world-to-camera extrinsic for frame $i$, $\mathbf{T}_{wo}$ is the world-to-object transformation, and $\mathbf{T}_{po}$ is the fixed part-to-object relationship. This hierarchical composition yields the rigid transformation of the part in the camera coordinate system for all frames, enabling consistent tracking and analysis.

### C.3. Model Inference and Qualitative Analysis.

We load the pre-trained weights (checkpoints) obtained from training to evaluate the model's generalization capability for unseen objects. The inference process consumes the processed real-world point clouds as input. For each frame $i$, the model predicts the 6D pose of the target part, outputting a unit quaternion $\hat{\mathbf{q}}^{(i)}$ representing the orientation and a translation vector $\hat{\mathbf{t}}^{(i)} \in \mathbb{R}^3$.

We present qualitative results for each object category, visualizing the predicted 6D poses as 3D bounding boxes projected onto the original RGB frames.

## D. More qualitative results on GAParts pose estimation

We demonstrate additional qualitative results for GAPart pose estimation as shown in Fig. D and Fig. D.

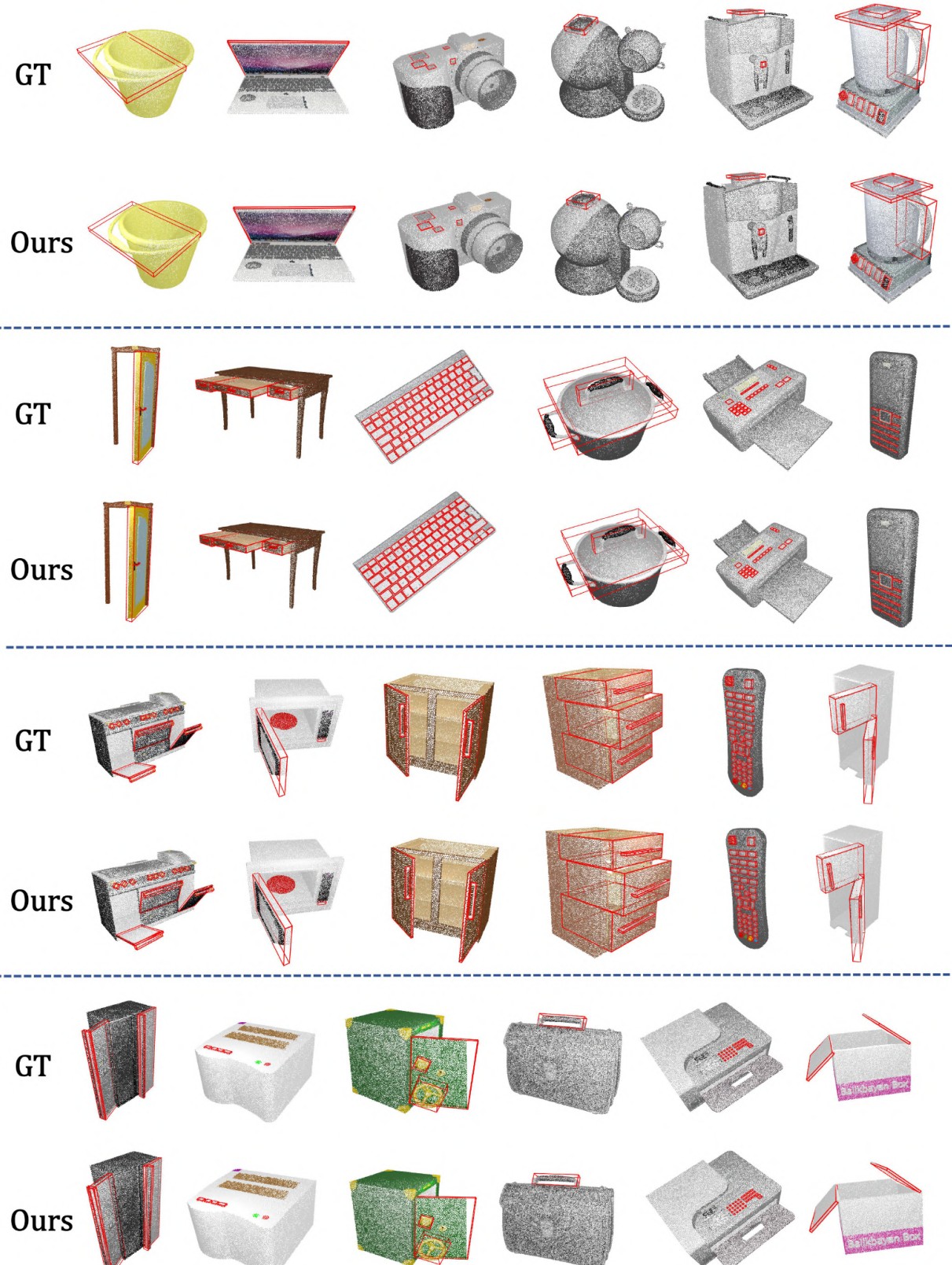

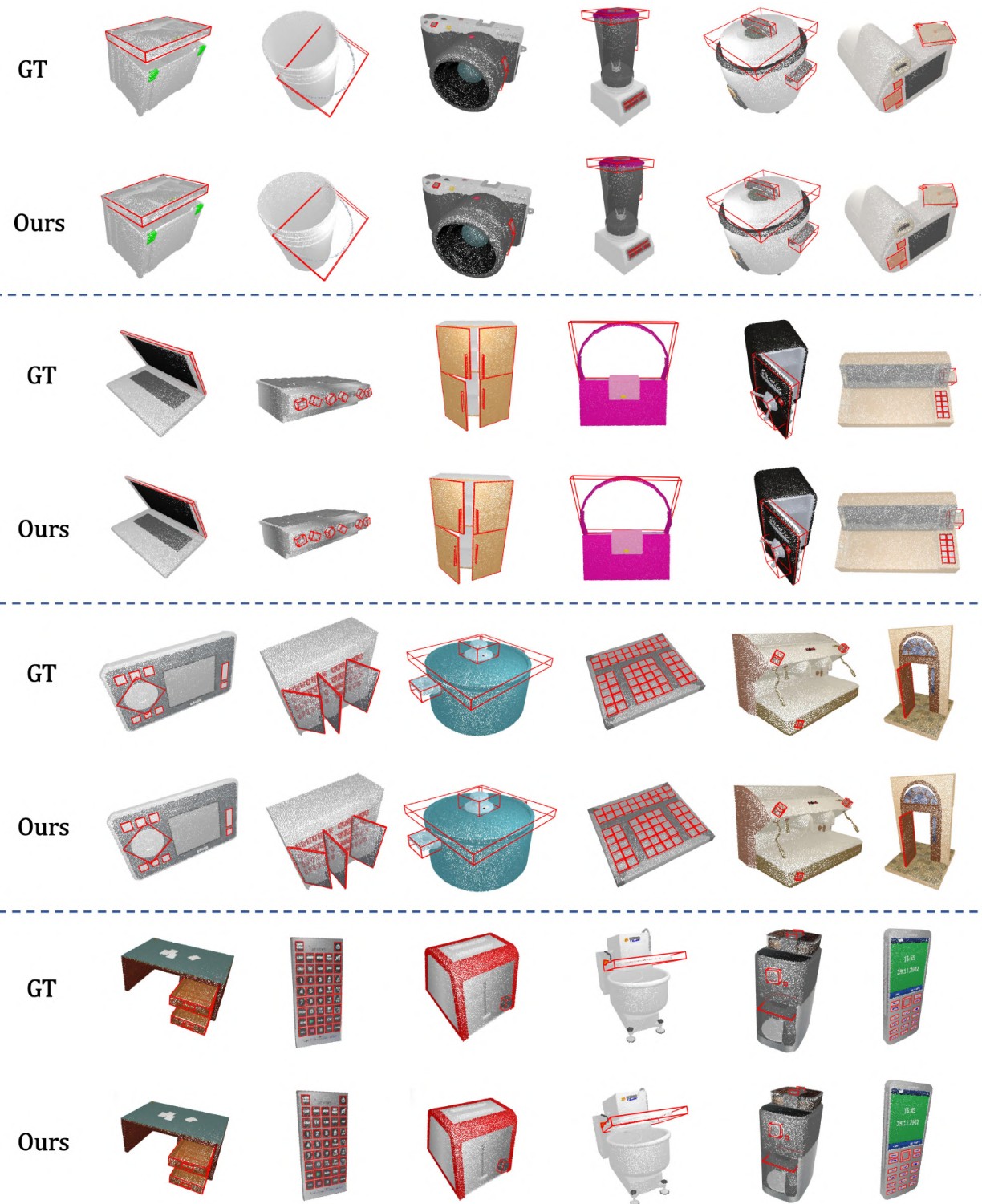

