# OpenReview forum: "Generalizable and Actionable Parts Pose Estimation with Symmetry Annotation-Free Learning Strategy"
_ICML.cc/2026/Conference — ICML 2026 regular_

### Official Review · Reviewer_ezUk · 2026-02-18

**Soundness:** 1
**Presentation:** 1
**Significance:** 3
**Originality:** 2
**Overall Recommendation:** 2
**Confidence:** 4

**Summary:**

The paper introduces a method for generalizable and actionable parts pose estimation, addressing the problem and dataset introduced in GAParts (Geng et al. 2023). The main contributions and improvement over the existing methods is the approach for rotation candidate generation which aims to estimate the distribution of part poses and assist in discovering the type of symmetry of the object.

**Compliance With Llm Reviewing Policy:**

Affirmed.

**Final Justification:**

Many thanks to the authors for their engagement during the rebuttal phase and the in-depth discussion. While some questions were clarified, I decided to keep my rating as is. The main reasons for this decision are:

- I find that the paper needs extensive rewriting and would thus benefit from an additional review process.
- The explanation and evaluation on how the candidate poses follow the real poses distribution for symmetric object is not sufficient in my opinion as discussed with the authors during the rebuttal interaction.

**Key Questions For Authors:**

I would be interested in some discussion on my last weakness point regarding the candidate rotation strategy (see above), as well as the fixed ref. vector t and the q,-q quaternion handling.

**Limitations:**

The paper does not provide an extensive discussion of limitations, or specific failure cases of the method.

**Strengths And Weaknesses:**

The addressed problem of part pose estimation is not yet extensively investigated but has significant importance for real-world robotic applications in unconstrained environments. The significance of research in the area is therefore high, and it is positive that the authors show an actual deployment of their method on a robotic arm. Furthermore, the problem of symmetry in object pose estimation is also very relevant, especially with respect to ground truth ambiguity in learning-based methods. Finally, the experimental evaluation is extensive and shows improvement over the baseline GAParts and other methods.

A key weakness area of the paper is the writing quality, organization and presentation. To be more precise and include specific examples:

- Sometimes the motivation for certain design decisions is unclear or the specific measure precedes its objective, making the paper very challenging to follow. For example, the motivation for using quaternions comes very late in the paper and the purpose of the candidate rotations generation only becomes clear very late.

- Expressions are used that do not fit scientific work, such as e.g. "enormous potential" in Line 33 (left), "seriously without doubt" in Line 32 (right), "tremendous potential" in Line 81 (left).

- Some abbreviations are introduced (e.g. NPCS) in Line 15 (right)) but not explained. On the same line, some dimensions are introduced that are unclear. For example, Line 096 (right) introduces labels from 1 to 9 but this due to the number of classes in GAPart, not a general problem definition.

- The related work section is limited in its coverage of the pose estimation problem as well as the approaches for symmetry handling in this field.

- One of the main claims of the paper is not requiring symmetry annotation, however later a need of pre-defining the type of symmetry is introduced.

- There are parts of the methodology that are not clearly described. For example, an introduction of a fixed reference vector t in Line 153 (left) - how is this vector defined and who fixes it, or the exact handling of the quaternion ambiguity q,-q (Line 215, left) - how does aligning the sign with the mean quaternion help here.

- (Minor) Line 438, Figure number is incorrect.

A key question regarding the methodology is how the candidate rotation generation method guarantees a successful estimation of the distribution of symmetric poses.

---

> ### Author Rebuttal · Authors · 2026-03-31
>
> **Q1 Clarification on Quaternion Sign Ambiguity**
>
> Regarding the quaternion ambiguity between q and −q, we would like to kindly clarify that this point has already been discussed in Sec. 3.3.2 (Candidates Refinement) of the main paper, where the alignment strategy is defined in Eq. (3). To be specific, since -q and q represent the same physical rotation on SO(3), we align the sign of each candidate according to the naive mean quaternion. After that, we compute the residual between each candidate and the mean quaternion
>
> **Q2 Concern on Annotation**
>
> We agree that our method is not completely annotation-free in the strictest sense, since the symmetry type is provided. However, this requirement is much weaker than manually annotating symmetry axes or symmetry planes. In practice, identifying the symmetry type can often be treated as a simple classification problem, and does not require the time-consuming geometric annotation needed for explicit symmetry axis or symmetry plane.
>
> **Q3 Candidates to Final**
>
> We thank the reviewer for the helpful question regarding the candidate-to-final prediction process. In the main paper, Fig. 2 already provides a qualitative analysis of the first-stage candidate estimation results. In addition, we further conduct a quantitative analysis here.
> Specifically, we conduct the evaluation on several selected categories from GAPart and analyze the first-stage candidates from two aspects:
> (1) the error between the generated candidates and the ground-truth pose (Cand vs GT);
> (2) the error between the generated candidates and the final predicted pose (Cand vs Final).
> For reference, we also report the error between the final prediction and the ground truth (GT vs Final).
>
> | Category | Setting | Cand vs GT | Cand vs Final | GT vs Final |
> |:-:|:-:|:-:|:-:|:-:|
> |Rd.F.HI|Seen|8.04|8.30|3.56|
> |Rd.F.HI|Unseen|8.60|8.79|3.94|
> |Sd.Ld|Seen|5.51|5.55|0.46|
> |Sd.Ld|Unseen|5.74|5.84|0.41|
> |Sd.Bn|Seen|6.78|6.85 |0.69|
> |Sd.Bn|Unseen|7.53|7.22|3.12|
> |Hg.Dr|Seen|15.39|15.08|1.00|
> |Hg.Dr|Unseen|15.73|15.70|3.40|
> |Hg.Ld|Seen|16.37|16.37|2.39|
> |Hg.Ld|Unseen|16.87|16.95|4.71|
>
> The results show that the generated candidates, the final prediction, and the ground-truth pose are all relatively close in the pose space, since Cand vs GT and Cand vs Final are highly consistent with only small differences. This indicates that the first-stage candidates already form a reasonable pose neighborhood. Moreover, the error of GT vs Final is further reduced, showing that aggregating the candidates can move the final prediction closer to the ground truth.
> However, since the final prediction is produced by aggregating the whole candidate set rather than selecting the oracle-best candidate, it is still affected by candidates with larger deviations and therefore cannot perfectly match the ground truth. Overall, these results verify that the candidate set is informative and that the second stage can further improve the prediction quality.
>
> We sincerely thank the reviewer for pointing out the issues related to writing, organization, and presentation. We fully acknowledge that the manuscript can be improved in these aspects, and we will carefully revise it to enhance clarity, refine inappropriate expressions, complete missing explanations of abbreviations and variables, and correct the formatting issues noted by the reviewer. At the same time, we have to emphasize that all these formatting issues do not have much influence on our manuscript's presentation quality, which may impede the reviewer from understanding the motivation and main technical contribution of our paper. We would appreciate it if you can consider and re-evaluate our manuscript's contribution objectively after correcting the formatting problems.
>
> Thank you again for your effort in the ICML review process. We are looking forward to further discussion with you!

---

> > ### Author Rebuttal · Reviewer_ezUk · 2026-04-01
> >
> > I would like to thank the authors for their extensive response and additional experiments. While this is appreciated and the work has merits, I would like to address the following with respect to their responses:
> >
> > Q1: Clear
> > Q2: I agree with the fact that this type of information on symmetry type is still advantageous with respect to symmetry axis annotation, the problem was that the introduction was creating a different expectation with respect to this, i.e. no symmetry annotation at all.
> > Q3: The provided result indicates that the candidate poses are good estimates of the pose in the mean sense. However, it does not provide information about the ability of the candidate generation to estimate a pose distribution for symmetric objects. Please clarify this point if you think there is a misunderstanding.
> >
> > Finally, the issues related to writing, organization and presentation quality are essential to a paper. The mentioned issues in my review were indicative, but I still think that the paper requires extensive rewriting.

---

> > > ### Author Response · Authors · 2026-04-01
> > >
> > > Dear reviewer, thank you for your further comments. We appreciate the opportunity to further clarify these points and will address the presentation concerns and supplementary questions below.
> > >
> > > **Regarding the Presentation Concern**
> > >
> > > We acknowledge that several unclear points about the presentation need to be further clarified. In order to make sure that you understand our modification schedule, we plan to modify our manuscript as outlined below, which can be accommodated according to the additional one-page allowance in the ICML camera-ready version:
> > >
> > > * We agree that putting the motivation behind the objective may cause a little confusion. Therefore, we acknowledge that we will thoroughly check these points and correct them. Specifically, we will bring forward the motivation for using quaternions to Sec. 3.1, "Overview", and the purpose of candidate rotation generation to Sec. 1, "Introduction", to make the presentation clearer.
> > >
> > > * We will thoroughly check expressions that do not fit scientific writing and correct them carefully. For example, we will replace "tremendous potential" with "enormous potential".
> > >
> > > * We apologize for the confusion caused by abbreviations (e.g., NPCS), and we promise that we will introduce the full explanation when they are first mentioned.
> > >
> > > * We will give a detailed description of the fixed reference vector t where it first appears. Specifically, we will supplement its definition in Sec. 3.2, "HyperS3 Convolution for Quaternion Regression", which we have already clearly explained in our response to reviewer dFsG in Q2.
> > >
> > > We would like to emphasize again that all these modifications mentioned above can be implemented according to the additional one-page allowance in the ICML camera-ready version. If our manuscript is accepted, we will definitely make these corrections.
> > >
> > > **Q2 Supplementary Explanation on Annotation**
> > >
> > > We have to clarify that GAPart Pose Estimation is originally a category-level part perception task. Since GAPart is classified by part category, the symmetry type can be treated as a category-level geometric prior rather than a precise symmetry annotation. Thus, the claim that "our method is annotation-free" is appropriate and not exaggerated, and it does not diminish our work's technical contribution or scientific value to the ICML community.
> > >
> > > **Q3 Supplementary Explanation on Candidates**
> > >
> > > We thank the reviewer for the opportunity to further clarify this point. As mentioned before, we provide a qualitative analysis of the candidates' distribution around the annotated ground-truth pose in the main paper, Fig. 2, which intuitively shows that the candidates are distributed around the equivalent solution distribution induced by symmetry. We explain the implicit reason below:
> > >
> > > We would like to clarify that the candidate set learns the pose distribution through the design of the supervision in the candidate generation stage. Specifically, during training, we do not use only the annotated ground-truth pose as the sole correct target. Instead, we treat the ground-truth pose together with its symmetry-equivalent poses (constructed based on the predicted symmetry information) as valid supervision targets. In this way, the model is encouraged to assign candidates to feasible poses within the equivalent solution set, rather than penalizing those symmetry-equivalent poses that are also correct. As a result, the generated candidates naturally distribute around the valid equivalent solutions, allowing the candidate set to capture the pose distribution associated with symmetric objects.
> > >
> > > We hope our clarification will address your concern. Thank you again for your effort in the ICML review process!

---

### Official Review · Reviewer_MMtu · 2026-02-18

**Soundness:** 2
**Presentation:** 3
**Significance:** 2
**Originality:** 2
**Overall Recommendation:** 2
**Confidence:** 4

**Summary:**

This paper presents a two-stage framework SAFAG, which adopts a symmetry annotation-free learning strategy for generalizable and actionable parts pose estimation. It first generates a set of quaternion candidates and then outputs the final quaternion. Extensive experiments on GAPartNet show that our approach achieves state-of-the-art performance.

**Compliance With Llm Reviewing Policy:**

Affirmed.

**Final Justification:**

Maintain my original score.

**Key Questions For Authors:**

1. My primary concern is the inference performance of this two-stage framework. In the actual deployment of VLA scenarios, the real-time requirement for intelligent agents is extremely high. Could you supplement the description with the approximate computational complexity or inference time of the network?
2. As a two-stage framework, could you quantitatively or qualitatively analyze the candidate estimation results of the first stage, so as to further demonstrate the process from candidates to determining the final quaternion?
3. Although the framework adopts a self-supervised approach, is such "self-supervision" necessary? Will the lack of explicit supervision lead to the framework's poor generalization on non-fully symmetric objects? Meanwhile, as a submission to a top academic conference, it would be better if the model could further prove its generalization ability on more datasets.

**My primary concern remains the real-time performance and generalization ability of this method. If the authors can prove the effectiveness of the framework, I will consider revising my rating. Best regards.**

**Limitations:**

The experimental settings are rather limited. The generalization ability under more diverse scenarios, especially for non‑fully symmetric objects, deserves further validation.

**Strengths And Weaknesses:**

**Strengths**

1. The motivation is easy to follow, and the visualization quality of the paper is high.
2. The experimental results show that the proposed method achieves state-of-the-art performance on GAPartNet.
3. Reproducible code is provided, and the code quality is high.

**Weaknesses**

See Questions.

---

> ### Author Rebuttal · Authors · 2026-03-31
>
> We thank the reviewer for the valuable comments and suggestions regarding the completeness of our experimental evaluation. If the paper is accepted, we will make full use of the additional one-page allowance in the ICML camera-ready version to include these supplementary experiments and further strengthen the empirical validation of our method.
>
>
> **Q1 Inference Time**
>
> We supplement a runtime comparison with several representative baselines. The results are shown below.
> |Method|FPS|
> |:-:|:-: |
> |GenPose|11.7|
> |GenPose++|8.7|
> |RFMPose|11.3|
> |DFGAP|28.38|
> |Ours|**74.19**|
>
> Our method achieves substantially higher inference speed than GenPose, GenPose++, and RFMPose. A main reason is that these methods are diffusion-based and require iterative denoising during inference, which leads to significantly higher inference cost. In addition, DFGAP is also slower because it relies on the pretrained DINO model during inference, which introduces extra computational overhead.
>
> **Q2 Candidates to Final**
>
> We thank the reviewer for the helpful question regarding the candidate-to-final prediction process. In the main paper, Fig. 2 already provides a qualitative analysis of the first-stage candidate estimation results. In addition, we further conduct a quantitative analysis here.
> Specifically, we conduct the evaluation on several selected categories from GAPart and analyze the first-stage candidates from two aspects:
> (1) the error between the generated candidates and the ground-truth pose (Cand vs GT);
> (2) the error between the generated candidates and the final predicted pose (Cand vs Final).
> For reference, we also report the error between the final prediction and the ground truth (GT vs Final).
>
> | Category | Setting | Cand vs GT | Cand vs Final | GT vs Final |
> |:-:|:-:|:-:|:-:|:-:|
> |Rd.F.HI|Seen|8.04|8.30|3.56|
> |Rd.F.HI|Unseen|8.60|8.79|3.94|
> |Sd.Ld|Seen|5.51|5.55|0.46|
> |Sd.Ld|Unseen|5.74|5.84|0.41|
> |Sd.Bn|Seen|6.78|6.85 |0.69|
> |Sd.Bn|Unseen|7.53|7.22|3.12|
> |Hg.Dr|Seen|15.39|15.08|1.00|
> |Hg.Dr|Unseen|15.73|15.70|3.40|
> |Hg.Ld|Seen|16.37|16.37|2.39|
> |Hg.Ld|Unseen|16.87|16.95|4.71|
>
> The results show that the generated candidates, the final prediction, and the ground-truth pose are all relatively close in the pose space, since Cand vs GT and Cand vs Final are highly consistent with only small differences. This indicates that the first-stage candidates already form a reasonable pose neighborhood. Moreover, the error of GT vs Final is further reduced, showing that aggregating the candidates can move the final prediction closer to the ground truth.
> However, since the final prediction is produced by aggregating the whole candidate set rather than selecting the oracle-best candidate, it is still affected by candidates with larger deviations and therefore cannot perfectly match the ground truth. Overall, these results verify that the candidate set is informative and that the second stage can further improve the prediction quality.
>
> **Q3 Generalization on another dataset**
>
> We thank the reviewer for the valuable suggestion on evaluating generalization beyond the GAPart Dataset.
> While GAPartNet itself is already a relatively comprehensive and representative benchmark for part-level perception and pose estimation, constructed from both PartNet-Mobility and AKB-48, we further evaluate our method on the ClearPose dataset to better assess its cross-dataset generalization.
> Specifically, we conduct experiments on the plate, water_cup, wine_cup, and bowl categories in ClearPose. These selected categories all exhibit clear symmetry properties, making them suitable for evaluating our symmetry-aware framework's generalization.
>
> For a fair comparison, we use the ground-truth depth provided by ClearPose to generate point clouds as input to our model.
>
> ||plate|water_cup|wine_cup|bowl|
> |:-:|:-:|:-:|:-:|:-:|
> |Rot.(°)|2.42|1.79|1.52|3.07|
> |Trans.(cm)|0.008|0.007|0.007|0.008      |
>
> These results show that our method remains effective on this non-part-centric dataset, suggesting that our framework has promising generalization ability beyond the GAPart Dataset.
>
> Thank you again for your effort in the ICML review process. We are looking forward to further discussion with you!

---

> > ### Author Rebuttal · Reviewer_MMtu · 2026-04-03
> >
> > Thank you for clarifying and explaining the details; I will take them into further consideration.

---

> > > ### Author Response · Authors · 2026-04-03
> > >
> > > Thank you so much for your reply! We're delighted to learn that we've successfully addressed your concerns.  We would truly appreciate it if you could kindly revise your rating.
> > >
> > > We will incorporate these additional experimental results and analyses into camera-ready version if accepted. If you have any further concerns, please don't hesitate to let us know.
> > >
> > > Once again, we extend our heartfelt gratitude for the tremendous effort you've put into reviewing our manuscript, as well as for the insightful and valuable suggestions you've offered. These suggestions have indeed been of great benefit to our work.

---

### Official Review · Reviewer_m1UN · 2026-02-24

**Soundness:** 4
**Presentation:** 3
**Significance:** 2
**Originality:** 2
**Overall Recommendation:** 4
**Confidence:** 4

**Summary:**

This paper proposes SAFAG, a two-stage quaternion regression framework to handle symmetry without requiring explicit symmetry-axis/plane annotations. The method generates multiple quaternion candidates, encodes their distribution in S³/tangent space, refines them, and aggregates to a final pose while a self-supervised “symmetry-aware” module estimates rotational axes or mirror planes to construct equivalent pose sets for training supervision.

**Compliance With Llm Reviewing Policy:**

Affirmed.

**Final Justification:**

My concerns have been addressed. I decide to raise my rating.

**Key Questions For Authors:**

1. How many quaternion candidates K and equivalent-set samples N are used, and how sensitive are results to these values?
2. How robust is the method to incorrect symmetry-type labels? Have you tested sensitivity to mis-specified type (rotational vs mirror vs none)?
3. How do you handle the non-differentiability of min operators in Eqs. (9–11)? Is there a soft-min or top-k relaxation to stabilize training?

**Limitations:**

Yes.

**Strengths And Weaknesses:**

Strengths:
1. The overall pipeline is clear and the paper is easy to follow.
2. The evaluations on GAPartNet is comprehensive and complete.

Weaknesses:
1. The rotational symmetry estimation appears weakly constrained: beyond using predicted axes to generate equivalent rotations, there is no explicit geometric or self-consistency term preventing degenerate axis predictions; the learning signal to $\pi_x, \pi_y, \pi_z$ is indirect and potentially circular.
2. Mirror symmetry is limited to three orthogonal candidate planes derived from predicted normals; real objects may exhibit non-orthogonal or multiple/plausibly continuous mirror structures, making the orthogonality assumption restrictive.
3. The method relies on knowing the symmetry type (rotational vs mirror vs none) per part class, so it is not fully annotation-free in practice.
4. Some important classes underperform DFGAP, e.g., unseen Hg.Kb, Hg.Dr and translation errors for certain unseen classes are large, e.g., 0.399 cm, but failure analyses are missing.
5. Comparisons appear incomplete. Recent single-stage GAParts baselines that fuse RGB-D or modern point backbones like CAP-Net are not included, which weakens the empirical claim of state of the art.
6. Several training/runtime details such as number of candidates K, equivalent-set sampling cardinality N, computational cost are missing.
7. The axis distribution as $\pi_x, \pi_y, \pi_z$ leading to $n= \pi_x n_x+ \pi_y n_y+ \pi_z n_z$ reads like a soft axis selection over canonical directions but in effect is just a linear combination. The probabilistic modeling motivation and normalization are unclear.

---

> ### Author Rebuttal · Authors · 2026-03-31
>
> We thank the reviewer for the valuable comments and suggestions regarding the need for clearer theoretical explanation and additional experimental support. We acknowledge that some theoretical details were not sufficiently elaborated in the current version. If the paper is accepted, we will make use of the additional one-page allowance in the ICML camera-ready version to further supplement both the theoretical analysis and the corresponding experiments.
>
> **Q1**
> Since the axis itself is not explicitly annotated, we supervise the symmetry axis indirectly in a self-supervised manner.
> However, the learning signal is not circular in practice. During the warm-up stage, the model does not rely on the predicted axis, but instead uses canonical axes to provide a guided initialization for subsequent axis learning.
> Once the predicted axis is introduced, the optimization is still anchored by the ground-truth pose, rather than being determined solely by the predicted axis itself.
> In practice, we find that this supervision is already sufficient for learning meaningful rotational axis without introducing an additional geometric consistency term. Avoiding such extra constraints helps keep the framework computationally efficient.
>
> **Q2**
> Since our three orthogonal candidate planes form an orthogonal basis, they provide a compact and structured representation for mirror-symmetry reasoning. More general mirror planes can then be expressed as combinations of the corresponding plane normals.
> Therefore, the orthogonality assumption mainly serves as a structured parameterization, rather than a strict restriction to mutually orthogonal mirror symmetries.
>
> **Q3**
> We agree that our method is not completely annotation-free in the strictest sense, since the symmetry type is provided. However, this requirement is much weaker than manually annotating symmetry axes or symmetry planes. In practice, identifying the symmetry type can often be treated as a simple classification problem, and does not require the time-consuming geometric annotation needed for explicit symmetry axis or symmetry plane.
>
> **Q4**
> From our analysis, these failure cases are mainly caused by severe occlusion in the dataset, where the target-part point clouds are heavily incomplete and therefore lack sufficient effective geometric information for reliable pose estimation. By contrast, DFGAP is an RGB-based method and can benefit more from category-level appearance and size priors. Such priors may help maintain stable translation estimates even when the visible geometric evidence is limited.
>
> **Q5**
> Thanks for your suggestion. We agree with the additional comparison and report the results below. The results of CAP-Net on GAPartNet dataset are captured from the original paper. We promise to include this comparison in the camera-ready version once the paper is accepted.
> |Method|Rot(°)|Trans(cm)|
> |:-:|:-: |:-:|
> |CAP-Net|10.39|0.055|
> |Ours|**7.03**|**0.041**|
>
> **Q7**
> Here, the probabilistic modeling is defined over a discrete latent axis variable rather than the full continuous space of 3D directions. Specifically, $\pi_x, \pi_y, \pi_z$ parameterize a discrete distribution over corresponding canonical axes.
> The predicted symmetry axis is then obtained as the normalized weighted sum of these canonical directions, which can be interpreted as the linear combination of these discrete axis distributions.
>
> **Q6 and Q8**
> Due to the space limitation of the response, for the detailed parameter settings, please refer to our response to dFsG's Q2, where we provide additional experimental analysis on several important hyperparameters. And for real-time performance experiment please refer to our response to MMtu's Q1.
>
> **Q9**
> We conduct additional experiments on rotationally symmetric objects by intentionally mislabeling their symmetry types as either mirror-symmetric or asymmetric, as shown in the table below.
> |Setting |Seen(°)|Unseen (°)|
> |:-:|:-:|:-:|
> |Correct rotational symmetry|3.35|9.11|
> |Mistaken as mirror symmetry|10.06|14.48|
> |Mistaken as asymmetric|25.01|28.90|
>
> The results show that when a rotationally symmetric object is treated as mirror-symmetric, the model can still capture part of the underlying symmetry and equivalent-set structure. In contrast, when it is treated as asymmetric, the rotational performance degrades noticeably, because the model cannot recognize the underlying symmetry and will incorrectly penalize pose predictions that are actually valid equivalent solutions.
>
> **Q10**
> We have indeed taken differentiability into consideration in our implementation. In our implementation, we address this issue using a soft-min relaxation.
> Specifically, we replace the original min operation with a temperature-controlled weighted average with β=10, which provides a smoother approximation and helps improve training stability.
>
> Thank you again for your effort in the ICML review process. We are looking forward to further discussion with you!

---

> > ### Author Rebuttal · Reviewer_m1UN · 2026-04-01
> >
> > My concerns have been addressed. I decide to raise my rating.

---

> > > ### Author Response · Authors · 2026-04-01
> > >
> > > We are delighted to know that your concerns have been fully addressed. Thank you for raising your rating!

---

### Official Review · Reviewer_dFsG · 2026-02-25

**Soundness:** 3
**Presentation:** 3
**Significance:** 4
**Originality:** 3
**Overall Recommendation:** 5
**Confidence:** 4

**Summary:**

This paper proposes SAFAG, an unsymmetrically labeled 6D pose estimation framework for generically applicable components (GAParts), which learns symmetrical structures through self-supervision and avoids relying on manually labeled symmetrical axes or planes. The method employs a multi-candidate quaternion generation and tangent space offset aggregation strategy to gradually refine the rotation estimation on the hypersphere, thereby effectively alleviating the multi-solution problem caused by symmetrical objects. Experiments on the GAPartNet dataset and real robot operations demonstrate that this method significantly improves pose accuracy and stability in both symmetrical and cross-category scenarios.

**Compliance With Llm Reviewing Policy:**

Affirmed.

**Final Justification:**

My concerns have been addressed. I decide to raise my rating.

**Key Questions For Authors:**

The author needs to provide more detailed explanations regarding the meanings of the formulas and variables.

**Limitations:**

The author could consider adding an experiment or analysis that is more closely aligned with real operational scenarios. For instance, when a single object contains both large-scale components (such as doors and drawers) and small-scale components (such as buttons and knobs), evaluate the impact of the proposed method on the selection of grasping strategies and the stability of pose estimation. By demonstrating the model's performance under different operational scales (overall operation vs. local operation) and combining simple strategy switching or grasping planning examples, it will help to more clearly illustrate the advantages and application scope of GAParts in actual robot operations, thereby further strengthening the motivation and application value of the paper.

**Strengths And Weaknesses:**

Strengths:
1 Symmetry annotation is not required to handle symmetric pose problems. This method explicitly models the rotational symmetry and mirror symmetry structures through self-supervised learning, avoiding the reliance on manually annotated symmetry axes or symmetry planes. This fundamentally alleviates the ambiguity problem of multiple solutions caused by symmetric objects, while also enhancing the generalization ability and practicality in scenarios with scarce data and new categories.
2 Design of a Stable Rotation Estimation Framework for Robot Operations. The paper proposes the generation of multiple candidate quaternions and The gradual refinement strategy of cutting space offset aggregation makes the rotation estimation more stable and robust, and can directly support downstream grasping and interaction tasks. The cross-category operation ability has been verified in real robot experiments, demonstrating the application value of the integration of perception and operation.
Weakness:
1 GAParts is an interesting concept. However, in reality, when robotic arms perform grasping, there are usually distinctions between large and small components, which determine whether the robotic arm grasps the target as a whole or in part. The author should clarify the scenarios where GAParts is applicable and its advantages, which is the motivation of the paper.
2 When describing the method, the author does not explain the settings of many parameters, such as: M nearest neighbors and fixed reference vector t, etc.
3 If v0 is a randomly sampled initialization vector, then is e1,i randomly generated each time? How can this possible random value be guaranteed to have a one-to-one correspondence with the target rotation? Also, e3,i = e1,i × e2,i, which makes the local rotation matrix contain only two parameters. What is the mathematical logic behind this?
4 The author uses tangent space offset regression in the quaternion space to refine the pose, but it is not clear what theoretical or empirical advantages this step has over direct quaternion regression or regression based on Lie algebra (SO(3) exponential map).

---

> ### Author Rebuttal · Authors · 2026-03-31
>
> We sincerely appreciate the reviewer's positive assessment of our paper. We thank the reviewer for recognizing the strengths of our work, including the insightful symmetry-aware learning strategy and the stable pose estimation framework. We are pleased to respond to your constructive comments and hope that our responses will clarify your concerns.
>
> **Q1 Clarification on GAParts**
>
> We thank the reviewer's insightful question regarding the applicability of GAParts.
> GAParts are designed for scenarios where robots interact with functionally consistent and reusable manipulation units that can be shared across different object categories.
> Under this definition, GAParts may correspond to either large-scale components (e.g., doors and drawers) or small-scale components (e.g., buttons and knobs), as long as these parts serve as reusable manipulation units with consistent interaction patterns. Accordingly, whether the robot operates on the object as a whole or on a local part depends on the task-relevant functional unit.
> The key advantage is that GAParts provide a unified representation for cross-category functional parts, enabling the model to generalize to unseen objects by leveraging shared functional structure.
>
> **Q2 Additional Details on Parameter Settings**
>
> We thank the reviewer for pointing out that several parameter settings were not sufficiently specified. We clarify the main settings as follows: the number of nearest neighbors is set to M=8; the reference vector is initialized to t=[1,0,0], and is switched to [0,1,0] when t is parallel to e1,i; the number of quaternion candidates is K=64; and the number of equivalent-set samples is N=36.
>
> We further conduct additional experiments on N and K to evaluate the sensitivity of our model to these parameters, and * denotes the setting used in our model.
>
> |K|Seen(°)|Unseen(°)|FPS|
> |:-:|:-:|:-:|:-:|
> |32|3.70|4.68|74.14|
> |*64|3.56|3.94|74.19|
> |128|3.23|3.99|73.07|
>
> |N|Seen(°)|Unseen(°)|FPS(with backpropagation)|
> |:-:|:-:|:-:|:-:|
> |18|5.97|11.77|6.84|
> |*36|3.56|3.94|4.33|
> |72|3.27|3.55|2.55|
>
> We note that the FPS reported for K is measured using only the forward pass, whereas the FPS reported for N is measured with backpropagation included since N is involved in the loss computation during training.
>
> For K, the model is not particularly sensitive to its value. Within a reasonable range, varying K leads to only minor changes in rotation performance, and K has a negligible impact on the computational time. This is because increasing K only expands the candidate dimension in the lightweight prediction head, while the computationally dominant backbone remains unchanged.
>
> For N, the model shows some sensitivity. A larger N provides denser supervision over the equivalent solution set and improves optimization performance, but it also increases the computational cost.
> This is because more sampled equivalent poses are involved in the loss computation during training. Therefore, we choose a moderate value of N to balance training efficiency and performance.
>
> **Q3 Clarification on the Local Canonical Frame**
>
> We appreciate the reviewer's question on the construction of the local canonical frame. v0 is introduced in the backbone stage only as an auxiliary initialization when constructing the local frame from the local covariance matrix Si.
> In practice, the resulting e1,i is mainly determined by the local geometric structure encoded in Si, while the effect of v0 is limited.
> Moreover, Ei is not used as a direct pose regression target. Instead, it serves as a local canonical frame for feature extraction in the SO(3) space. Thus, v0 does not need to have a one-to-one correspondence with the target rotation.
> Moreover, Ei is constrained to be a right-handed orthonormal basis with
> det(Ei)=1, which is consistent with a valid rotation matrix in SO(3). Based on the mathematical properties of the rotation matrix, the three vectors constituting the rotation matrix should be mutually orthogonal. Thus, e3,i can be uniquely fixed by e1,i and e2,i.
>
> **Q4 Motivation for Using Tangent Space**
>
> The refinement stage converts global rotation prediction into local residual estimation, which is generally easier to optimize than directly regressing the full target rotation from scratch.
> In practice, we adopt quaternion tangent-space updates because they are geometrically consistent and naturally aligned with our quaternion-based rotation representation in the overall pipeline.
> The refinement corresponds to updating the rotation along a local direction in the tangent space, which can be treated as a first-order variation on the manifold. Therefore, performing updates in the quaternion tangent space provides a clear geometric interpretation and can be viewed as following a gradient-like direction on the manifold.
>
> Thank you again for your effort in the ICML review process. We are looking forward to further discussion with you!

---

> > ### Author Rebuttal · Reviewer_dFsG · 2026-04-04
> >
> > My concerns have been addressed. I decide to raise my rating.

---

> > > ### Author Response · Authors · 2026-04-04
> > >
> > > Thank you so much for your response and for raising your rating! We're delighted to learn that we've successfully addressed your concerns. If you have any further concerns, please don't hesitate to let us know.

---

### Decision · Program_Chairs · 2026-04-30

**Decision:**

Accept (regular)

**Comment:**

Overall, the reviewers are positive about the important problem tackled and provided solutions and experimental supports. The author rebuttals have successfully addressed their initial concerns, and the reviewers decided to increase ratings. ACs note the reject recommendation of MMtu is inconsistent with his or her reviews, where the reviewer said the concerns have been adequately addressed by the rebuttals. The main concern was about the efficiency and the authors report much higher FPS than other methods. The reviewer ezUk mainly complains about presentations and clarities, which are also shared with dFsG who asked more detailed explanations on the meanings of the formulas and variables. dFsG, however, gave a clear accept recommendation. Given the positive reviews, and rather the extreme low ratings not equipped with strong justifications, ACs think the clarity issues are fixable and recommend the authors go through the paper, improving the presentation according to ezUk and dFsG.